# Improved global sea surface height and currents maps from remote sensing and in situ observations

Maxime Ballarotta[1], Clément Ubelmann[2], Pierre Veillard[1], Pierre Prandi[1], Hélène Etienne[1], Sandrine Mulet[1], Yannice Faugère[1], Gérald Dibarboure[3], Rosemary Morrow[4] & Nicolas Picot[3]

[1]Collecte Localisation Satellites, 31520 Ramonville-Saint-Agne, France
[2]Datlas, 38400 Saint Martin d'Hères, France
[3]Centre National d'Études Spatiales, 31400 Toulouse, France
[4]Centre de Topographie des Océans et de l'Hydrosphère, Laboratoire d'Etudes en Géophysique et Océanographie Spatiale, CNRS, CNES, IRD, Université Toulouse III, Toulouse, France

*Corresponding author: M.Ballarotta (mballarotta@groupcls.com)*

**Abstract.**

**We present a new gridded sea surface height and current dataset produced by combining observations from nadir altimeters and drifting buoys. This product is based on a multiscale & multivariate mapping approach that offers the possibility to improve the physical content of gridded products by combining the data from various platforms and in resolving a broader spectrum of ocean surface dynamic than in the current operational mapping system. The dataset covers the entire global ocean and spans from 2016-07-01 to 2020-06-30. The multiscale approach decomposes the observed signal into different physical contributions. In the present study, we simultaneously estimate the mesoscale ocean circulations as well as part of the equatorial wave dynamics (e.g., tropical instability and Poincaré waves). The multivariate approach is able to exploit the geostrophic signature resulting from the synergy of altimetry and drifter observations. Sea level observations in Arctic leads are also used in the merging to improve the surface circulation in this poorly mapped region. A quality assessment of this new product is proposed with regard to an operational product distributed in the Copernicus Marine Service. We show that the multiscale & multivariate mapping approach offers promising perspectives for reconstructing the ocean surface circulation: leads observations contribute to improve the coverage in delivering gap free maps in the Arctic; drifters observations help to refine the mapping in regions of intense dynamics where the temporal sampling must be accurate enough to properly map the rapid mesoscale dynamics; overall, the geostrophic circulation is better mapped in the new product, with mapping errors significantly reduced in regions of high variability and in the equatorial band; the resolved scales of this new product are therefore between 5% and 10% finer than the Copernicus product.**

## 1 Introduction

Several oceanographic applications (e.g., operational oceanography, marine weather, climate monitoring…) rely on high-quality observational datasets. The European Union (E.U.) Copernicus Marine & Climate Change Services provide operational services and indicators on the observed state of the climate. Sea level and surface currents are, among others, key variables distributed by the services. There are also listed as Essential Climate Variables (ECVs) for the detection of climate change and the characterization of climate system variability (Bojinski et al., 2014).

As part of the Copernicus Services, the Sea Level Thematic Assembly Centre (SL-TAC) delivers near-real time and delayed time sea level and surface currents products (along-track Level-3 and gridded Level-4 products) that are used by the ocean science community to study, understand and monitor the evolution of the ocean system. These products do not resolve the entire spectrum of the ocean surface variability; they have resolution limits of about 60 km for the along-track products (Dufau et al., 2016) and >200 km x 20 days for the gridded products (Ballarotta et al., 2019), but recent nadir altimetry instruments, such as the new Sentinel-3A and 3B SAR missions, or future missions based on large swath technologies (e.g., the upcoming Surface Water and Ocean Topography SWOT mission) offer, for example, the possibility of observing finer ocean structures (Morrow et al., 2019) which could be used to provide better gridded product resolution.

In addition, the growing needs to develop observing systems or methods with finer spatial scales / higher frequencies have been identified by the ocean scientific community and the Copernicus Services as R&D priorities to serve Copernicus marine users and decision-makers (see, e.g., Abdalla et al., 2021, or the "Copernicus Marine Service Evolution Strategy: R&D priorities - Version 5 June 30, 2021" document, https://marine.copernicus.eu/sites/default/files/media/pdf/2021-09/CMEMS%20Service_evolution_strategy_RD_priorities_v5-June-2021.pdf, last-access: 20221201). Therefore, with the support of the French Space Agency (CNES), the development of new experimental products has been undertaken, aiming at improving the resolution of the current Level-3 and Level-4 Sea level products (Mulet et al. 2021a, Ballarotta et al., 2020, Ubelmann et al., 2022, Prandi et al., 2021) and preparing operational systems for the SWOT era (Ubelmann et al., 2015, Ubelmann et al., 2021, Le Guillou et al., 2021, Beauchamp et al., 2020).

The present study focuses on the development and assessment of experimental global gridded products based on a recent multiscale & multivariate mapping approach (Ubelmann et al., 2021, 2022) and applied to real Earth observations. We here investigate the possibility of improving the content of gridded products in combining the data from various platforms (in situ and satellite) and in resolving a larger spectrum of the ocean surface dynamic than in current operational products.

The paper is structured as follows: the data sources and merging methods used in this study are described in section 2. Section 3 presents the experiments and validation metrics. The quality assessment of the new products is proposed in section 4. The key results are then summarized in section 5.

## 2 Data & Methods

### 2.1 Data sources

The mapping method used in this study takes input data from remote sensing and in situ observations, which are summarized
in Table 1 and described below.

**Table 1: List of observation datasets used in this study**

| Product type | Global Altimeter SLA products | Arctic leads Altimeter SLA products | Drifters' geostrophic velocity product |
|---|---|---|---|
| Product ref. | SEALEVEL_GLO_PHY_L3_REP_OBSERVATIONS_008_062 | Experimental | AOML |
| Spatial coverage | [0°E:360°E] [90°S:90°N] | >60°N | [0°E:360°E] [90°S:90°N] |
| period | From 2016-01-15 to 2020-06-30 | From 2016-01-15 to 2020-06-30 | From 2016-01-15 to 2020-06-30 |

### 2.1.1 Sea level anomaly products

The global ocean Sea Surface Height (SSH) observations are from the (Delayed-Time DT) Level-3 altimeter satellite along-
track data, reprocessed in 2021 and distributed by the E.U. Copernicus Marine Service (product reference
SEALEVEL_GLO_PHY_L3_MY_008_062, https://doi.org/10.48670/moi-00146). These data cover the period from 1993-
01-01 to 2020-12-31 over the world ocean (excluding ice-covered areas, e.g., Figure 1) and are available at a sampling rate
of 1 Hz (~7 km spatial spacing). Homogenisation and cross-validation are applied to the dataset to remove any residual orbit
error, long-wavelength error (lwe), large-scale biases and discrepancies between different data streams. The list of
geophysical and environmental corrections applied to the datasets is described in the Quality Information Document
(Taburet et al., 2021) and summarized below in Equation (1). In this study, unfiltered sea level anomalies (SLA) corrected
with dynamic atmospheric correction (dac), ocean tide and lwe corrections are considered in the multi-scale & multivariate
mapping.

$$\text{SLA} = \text{Orbit} - \text{Range} - \sum(\text{Environmental Corrections}) - \sum(\text{Geophysical Corrections}) - \text{Mean Sea Surface} \qquad (1)$$

with $\sum$ (Environmental Corrections) = wet tropospheric + dry tropospheric + ionospheric + sea-state-bias, $\sum$ (Geophysical
Corrections) = solid earth tide + load tide + ocean tide + pole tide + dynamic atmospheric correction (see Taburet et al.,
2021, for the references associated to each mission corrections). The Mean Sea Surface used here is the CNES-CLS18
(Mulet et al., 2021b).

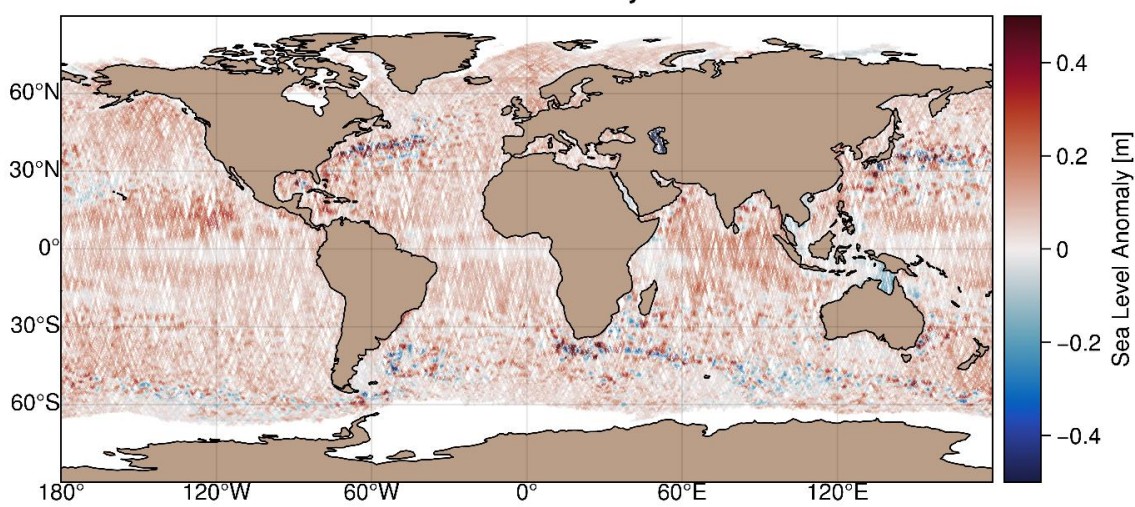

**Figure 1: Example of sea level altimetry coverage for a 7-day period (from 2019-07-01 to 2019-07-07). Colour scale represents the sea level anomaly amplitude in meters. For this time interval, data originate from six altimeters: Jason-3, Sentinel-3A, Sentinel-3B, SARAL/Altika, Cryosat-2, Haiyang-2A**

**2.1.2 Sea level anomaly products in artic leads**

In the polar regions satellite sea level observations are limited by the sea ice. Thanks to a dedicated processing, sea level can however be estimated within fractures in the ice (leads). The echoes from the altimeters over the ice-covered region are classified to identify peaky waveforms corresponding to lead echoes. Range estimation is then made with specific retracking methods, and it is corrected from instrumental and geophysical corrections to get sea level anomaly (Prandi et al. 2021). To

ensure continuity with the open ocean, the corrections are derived from the global ocean Level-3 along-track processing (Taburet et al., 2021) when possible. The noticeable exceptions concern 1) the wet tropospheric correction that comes from the European Centre for Medium-Range Weather Forecasts (ECMWF) model since on-board radiometer estimates are not reliable over ice, 2) the sea state bias correction is not applied since waves and winds are considered small over leads, 3) orbit error corrections are not applied as they are difficult to compute over this small region. Then a constant bias of ~8cm is

applied for each mission to ensure continuity with the SEALEVEL_GLO_PHY_L3_MY_008_062 open ocean SLA previously described. These products cover the Arctic region (up to 88°N) at a sampling rate of 20 Hz (~350 metres) for 3 altimetry missions: SARAL/AltiKa, Sentinel-3A and CryoSat-2 (Figure 2 & Table 2).

**Table 2: Arctic leads product characteristics**

| Altimeter | SARAL/Altika | Sentinel-3A | CryoSat-2 |
|---|---|---|---|
| Latitude max. | 81,5°N | 81,5°N | 88°N |
| Retracking | Adaptive (LRM) | TFMRA 50% (SAR) | TFMRA 50% (SAR) |


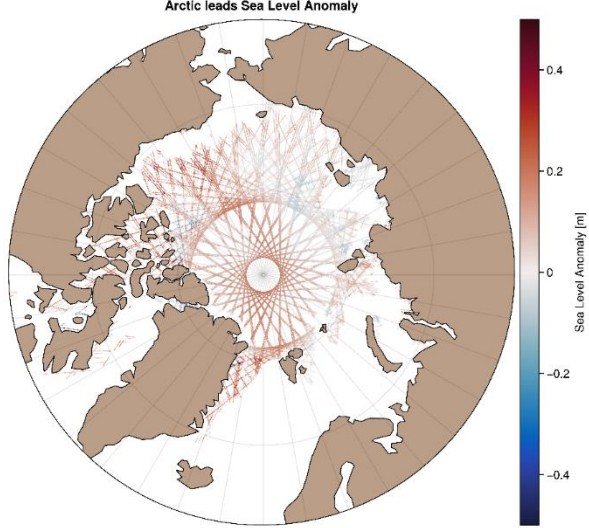

**Figure 2: Example of arctic leads sea level altimetry coverage for a 7-day period (from 2019-07-01 to 2019-07-07). Colour scale represents the sea level anomaly amplitude in meters**

### 2.1.3 Geostrophic current anomaly products

To further constrain the surface circulation, we used delayed-time horizontal surface velocities from the NOAA's Atlantic Oceanographic and Meteorological Laboratory (AOML) Surface Velocity Program (SVP, Lumpkin and Centurioni, 2019). The data cover the entire world ocean and are available at a 6-hour frequency. SVP are designed to follow the 15 m depth circulation, which is the centre depth of their drogues. When the drogue is lost, they follow the surface current, but are also under the direct influence of the wind. AOML distributes a flag to indicate whether the drogue is lost or not (Lumpkin et al, 2013). These data are also distributed by the INSITU Thematic Assembly Centre of the E.U. Copernicus Marine Service (see Product User Manual, http://marine.copernicus.eu/documents/PUM/CMEMS-INS-PUM-013-044.pdf) with an additional wind slippage correction for undrogued buoys derived from the Rio (2012) methodology. For the study, the undrogged and drogued drifters are selected over the global ocean and the period from 2016-06-01 to 2020-07-31. Note that for specific experiments described hereafter, we excluded drifters' trajectories between -10°S and 10°N (e.g., Figure 3) to isolate and evaluate only the impact of the equatorial wave's mode in this region. As in Mulet et al. (2021a), we computed the geostrophic velocity anomaly components, which are defined as:

$$U_{\text{anom}} = U_{\text{buoy}} - U_{\text{ekman}} - U_{\text{stokes}} - U_{\text{inertial}} - U_{\text{tidal}} - U_{\text{ahf}} - U_{\text{slip}} - U_{\text{mdt}} \qquad (2)$$

$$V_{\text{anom}} = V_{\text{buoy}} - V_{\text{ekman}} - V_{\text{stokes}} - V_{\text{inertial}} - V_{\text{tidal}} - V_{\text{ahf}} - V_{\text{slip}} - V_{\text{mdt}} \qquad (3)$$

With $U_{buoy}$ ($V_{buoy}$) is the drifter's zonal (meridional) velocity. Each component is corrected from:

- the wind-driven component $U_{ekman}$ ($V_{ekman}$) using an update of the model used in Mulet et al. (2021a) and described in Etienne (2021). The Ekman component is not available in the Mediterranean basin, so there is not drifter used in this region for the study. In this recent version, ERA5 wind stress (Hersbach et al, 2018) replaces the ERAinterim data and the equatorial symmetry of the wind driven parameters is removed.

- The Stokes drift $U_{stokes}$ ($V_{stokes}$) from ERA5 reanalysis (Hersbach et al, 2018) is also removed from the surface drifter velocity (undrogued drifters). No Stokes drift is removed from the 15m depth velocity, as this component is supposed to mostly vanish in the first 2-4 m.

- The wind slippage which is the direct effect of the wind on the buoy $U_{slip}$ ($V_{slip}$). This correction is significant only in the case of drogue loss (Etienne et al, 2021), when the drifters are advected by the surface current.

Then the data is filtered from the tidal and inertial velocities $U_{inertial}+U_{tidal}$ ($V_{inertial}+V_{tidal}$) as well as the residual high frequency ageostrophic signal $U_{ahf}$ ($V_{ahf}$). Finally, the mean geostrophic velocity (CNES-CLS2018, Mulet et al., 2021b) $U_{mdt}$ ($V_{mdt}$) is subtracted to obtain the geostrophic velocity anomaly.

**Drifters velocity**

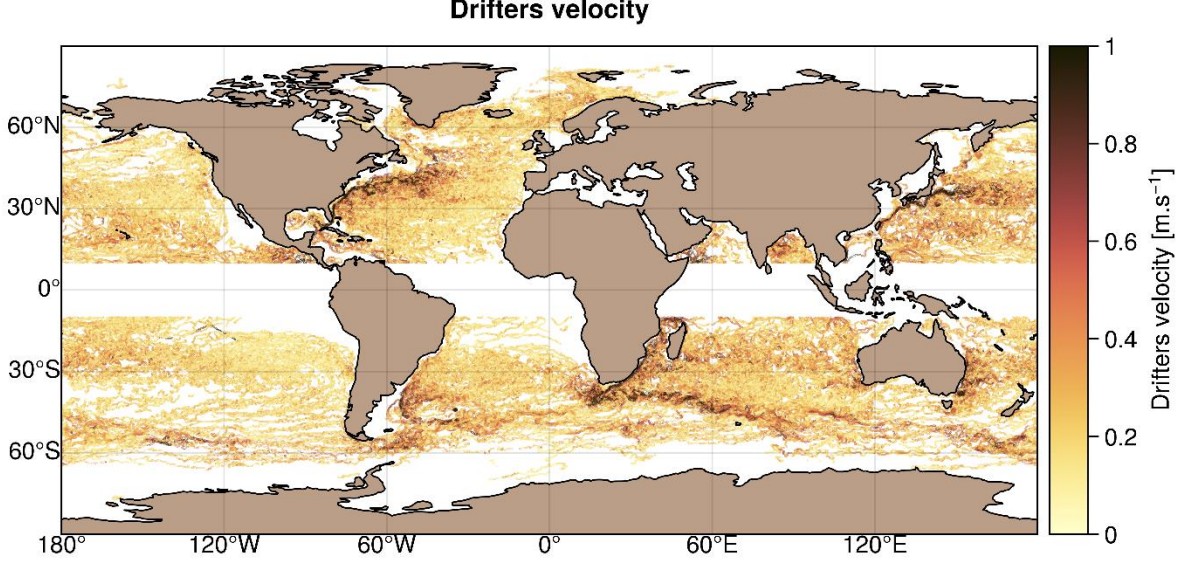

**Figure 3: Example of drifter's trajectories coverage for the 2019-01-01 to 2019-12-31 period. Colour scale represents the velocity**
**amplitude in m.s$^{-1}$**

### 2.2 Methods

Two mapping methods are compared in this study: the operational DUACS (Data Unification and Altimeter Combination System) mapping approach and the multiscale & multivariate MIOST (Multiscale Inversion of Ocean Surface Topography)

mapping approach. Each method is described in detail in reference articles, such as Le Traon et al (1998, 2003), Ducet et al. (2000) or Pujol et al. (2016) for the DUACS method; and Ubelmann et al. (2021, 2022) for the MIOST method. A description of the methods is given in Appendix A, and we propose hereafter to focus on the specific developments and processes that are considered in this study.

It is important to mention that DUACS maps are constrained by a single scale covariance function (Ahran and Colin de Verdière, 1985; Le Traon et al., 1998) and focus mainly on the geostrophic circulation (i.e., processes with typical space and time scales > 100km, 10 days). Consequently, they do not resolve the full spectrum of ocean surface variability. It is for example the case for the equatorial surface dynamics (see, e.g., Figure 7). While slow Rossby waves are already resolved within geostrophy in DUACS maps, faster equatorial waves such as Poincaré waves are filtered out, even though the space-time coverage of altimetry data allows sampling of large-scale waves with periods of 4-10 days and more (Farrar and Durland, 2022). The multi-scale approach proposed by the MIOST method offers the possibility to solve some of the missing surface variabilities in DUACS, accounting for the covariances of various surface processes in a single inversion. The covariance functions in the MIOST system are expressed as wavelet modes and the inversion is performed in this space using a variational approach (Ubelmann et al, 2021). In the following, we focus on the main components that have been tested in this study with the MIOST method: the geostrophy component already investigated in Ubelmann et al. (2021) and two new components associated to the equatorial wave's dynamic.

The geostrophy component follows the same formulation provided in Ubelmann et al. (2021) (see, their section 2.3.2.1, where the analytical formula of the ensemble of wavelet elements is given) and is also reported in Appendix A. The covariance function associated to the geostrophy component is plotted for a given point (210°E, 5°N) on Figure 4a and 4c, shown as a function of space (bottom left panel) and as a function of time (top left panel). This covariance function is similar to what is currently used for altimetry mapping with DUACS.

In the present study, we simultaneously estimate the surface signatures of the geostrophy, and equatorial Tropical Instability Waves (TIW) and Poincaré waves. As for the geostrophy component, the equatorial waves covariances are expressed as a reduced wavelet basis with typical wavelength and propagation speed given in the literature (e.g., Shinoda et al., 2009; Farrar, 2008, 2011; Farrar and Durland, 2012; Tanaka and Hibiya, 2019). For Poincaré waves, we built an ensemble of wavelet between 10°S and 10°N which follow the dispersion relation (Matsuno, 1966):

$$\omega = \sqrt{k^2 . c^2 + \beta . c \ . (2. n + 1)} \qquad (4)$$

where $\omega$ is the time frequency, $c = \pm 2.8 \ m. s^{-1}$ is the Poincaré waves propagation speed (considered as a constant here), $k$ the spatial wavenumber and $n$ a positive integer defining the waves mode. The wavelets are localized with a Hamming

window having half-widths of 1000 km in the zonal direction, 300 km in the meridional direction and 5 days in the temporal direction. For the TIW component, we also built an ensemble of wavelet between 10°S and 10°N which follow the dispersion relation (Matsuno, 1966):

$$\omega = c \ . k \tag{5}$$

where $\omega$ is the time frequency, $c = -0.5 \ m.s^{-1}$ is the TIW propagation speed (considered as a constant here), and $k$ the spatial wavenumber. The wavelets are here localized with a Hamming window having half-widths of 500 km in the zonal direction, 300 km in the meridional direction and 20 days in the temporal direction. The covariance function for a westward propagation wave like TIW is illustrated on Figure 4b and 4d for a given point (210°E, 5°N), shown as a function of space (bottom right panel) and as a function of time (top right panel). Note that for Poincaré waves, both eastward and westward propagation are considered. A more detailed description of the equatorial wave's components implemented in MIOST is provided in the Appendix A.

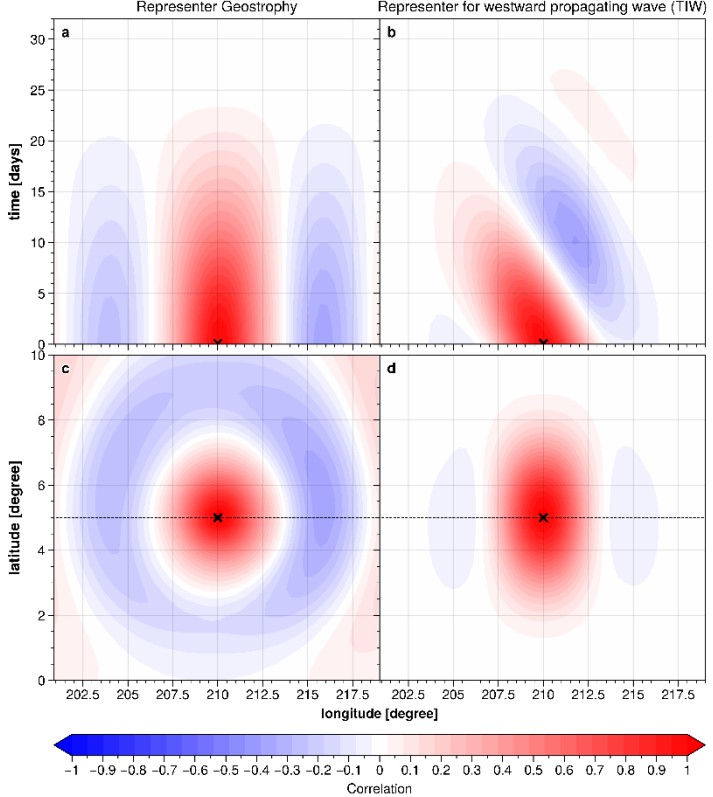

Figure 4: Example of spatio-temporal covariance models at (210°E, 5°N) for a), c) the geostrophy component (left panel) and for b), d) a westward propagating wave component, e.g., TIW (right panel)

## 3 Experiments and validation metrics

### 3.1 Experiments

We produced 4 years (from 2016-07-01 to 2020-06-30) of SSH maps using the MIOST multiscale & multivariate approach by combining the Level-3 altimeter dataset from SARAL/Altika, Envisat, Jason-1, Jason-2, Jason-3, Cryosat-2, Haiyang-2A, Haiyang-2B, Sentinel-3A, Sentinel-3B missions, the Level-3 arctic leads sea-level anomaly products from SARAL/AltiKa, Sentinel-3A and CryoSat-2 missions and geostrophic current anomaly data from AOML drifter database. These MIOST products are available on the AVISO+ (Archivage, Validation et Interprétation des données des Satellites Océanographiques) website (see section 6: Data availability, for more details).

Specific maps were also made to quantitatively assess the quality of these MIOST products. Table 3 summarises the list of experiments conducted in this study, indicating the input data used in the mapping and the physical content of the maps.
*DUACS allsat-1* and *MIOST allsat-1* experiments focus on the geostrophic variability. These SSH maps were produced from six altimeters (Jason-3, Cryosat-2, Sentinel-3A, Sentinel-3B, Haiyang-2A, Haiyang-2B) for the period 2019-01-01 to 2019-12-31, excluding one altimeter (Saral/Altika, *over open ocean region*) from the mapping to perform independent assessments. The *MIOST allsat-1 80% drifters + equatorial waves+ L3 arctic* experiment focuses on the geostrophic and equatorial waves variabilities. This experiment is based on 1) 80% of the drifter data, 2) the six altimeters previously mentioned over ocean and 3) leads altimeter observations. The Saral/Altika dataset (*over open ocean region*) and the remaining 20% of the drifter trajectories were here excluded from the mapping to perform independent assessments. Note that for these specific maps, drifter trajectories between -10°S and 10°N (e.g., Figure 3) were also excluded to evaluate only the impact of the equatorial wave's mode in this region.

**Table 3: List of mapping experiments with the input data and physical content considered**

| | Input data | | | Physical content | |
|---|---|---|---|---|---|
| Experiment | altimeter | drifters | L3 arctic | geostrophy | equatorial waves |
| DUACS allsat-1 | All w/o Altika | No | No | Yes | No |
| MIOST allsat-1 | All w/o Altika | No | No | Yes | No |
| MIOST allsat-1 80% drifters + equatorial waves+ L3 arctic | All w/o Altika | Yes (80%) | Yes | Yes | Yes |

### 3.2 Validations metrics

The validation metrics are based on statistical and spectral analysis.
One quantitative assessment is based on the comparison between SSH maps and independent SSH along-track data. This diagnostic follows 3 main steps: 1) the SSH gridded data is interpolated to the locations of the independent SSH along-track, geo-referenced by their longitude, latitude, and time; 2) the difference $SSH_{error} = SSH_{map-} SSH_{alongtrack}$ is calculated and 3) a

statistical analysis on the SSH$_{error}$ is performed in $1° \times 1°$ longitude $\times$ latitude boxes. Prior to the statistical analysis, a filtering operation can be applied to isolate the spatial scales of interest. For example, the analysis can be performed over the spatial range [65 km:500 km] typically representative of the medium mesoscale ocean signal. This excludes the noisy part of the reference signal (along-track) as well as possible large-scale biases (scale > 500 km). In the study, the validation metric is based on the error variance scores in $1°x1°$ longitude x latitude boxes (or averaged over specific region of interest), defined as:

$$\sigma_{err}(x, y) = \frac{\sum_{t=1}^{N}\left(SSH_{error}(x,y,t) - \overline{SSH_{error}(x,y,t)}\right)^2}{N} \tag{4}$$

The similar statistical analysis can also be performed on the geostrophic velocity errors U$_{error}$ = U$_{map}$- U$_{drifter}$, for the zonal component, and V$_{error}$ = V$_{map}$ - V$_{drifter}$, for the meridional component.

The comparison of the error variance score between two experiments informs about the gain or reduction Δ of the mapping error, for example:

$$\Delta = 100.\frac{\sigma_{err}(EXP2) - \sigma_{err}(EXP1)}{\sigma_{err}(EXP1)} \tag{5}$$

The previous diagnosis is undertaken in physical space (space/time space). For a more descriptive assessment by wavelength and to avoid spatio-temporal filtering of independent and study datasets, diagnostics can be performed in frequency space, using spectral analysis of SSH altimetry and gridded datasets. More specifically, a spectral analysis can be applied to altimetry data to estimate the effective resolution of gridded SSH products. It is described for example in Ballarotta et al. (2019). Here, we recall the main processing steps for the estimation of the effective resolution: 1) the SSH$_{map}$ data are interpolated to the locations of independent SSH$_{alongtrack}$ data, 2) the along-track and interpolated data are divided into overlapping segments of 1500 km length every 300 km, 3) each segment is stored in a database and referenced by its median coordinates (longitude, latitude), 4) finally, between latitudes 90°N-90°S and longitudes 0°-360°E, we consider $10° \times 10°$ longitude $\times$ latitude boxes for the global products every 1° increment. All available segments referenced in the $10° \times 10°$ box are selected to compute the mean power spectral densities of the independent signal (SSH$_{alongtrack}$) and the mapping error (SSH$_{map}$- SSH$_{alongtrack}$). Before the spectral calculation, the signals are detrended and a Hanning window is applied. The signal-to-noise ratio (Equation 6) is then derived from the power spectral density of the PSD along the trace (SSH$_{alongtrack}$) and the power spectral density of the error (SSH$_{map}$- SSH$_{alongtrack}$). As in Ballarotta et al. (2019), the effective resolution is then given by the wavelength λs where the SNR(λs) is 2 (Equation 7), i.e., the wavelength where the SSH$_{error}$ is two times lower than the signal SSH$_{alongtrack}$.

$$SNR(\lambda) = \frac{PSD(SSH_{along-track})(\lambda))}{PSD(SSH_{error})(\lambda)} \tag{6}$$

$$SNR(\lambda s) = 2 \tag{7}$$

## 4 Results

### 4.1 Qualitative assessment

We here qualitatively assess the gridded products from the *DUACS allsat-1* and *MIOST allsat-1 80% drifters + equatorial waves+ L3 arctic* experiments. The SLA maps from the DUACS and MIOST mapping approaches are relatively similar in the subpolar region, as illustrated in Figure 5 by an example of SLA reconstruction on 2019-02-15 for a) the DUACS mapping approach and b) the MIOST mapping approach. More significant differences take place in the Arctic basin: in contrast to the DUACS products, the use of arctic leads observations in MIOST offers the possibility to extend sea level mapping into ice-covered area and thus to deliver gap-free maps to the end-users (Figure 5b).

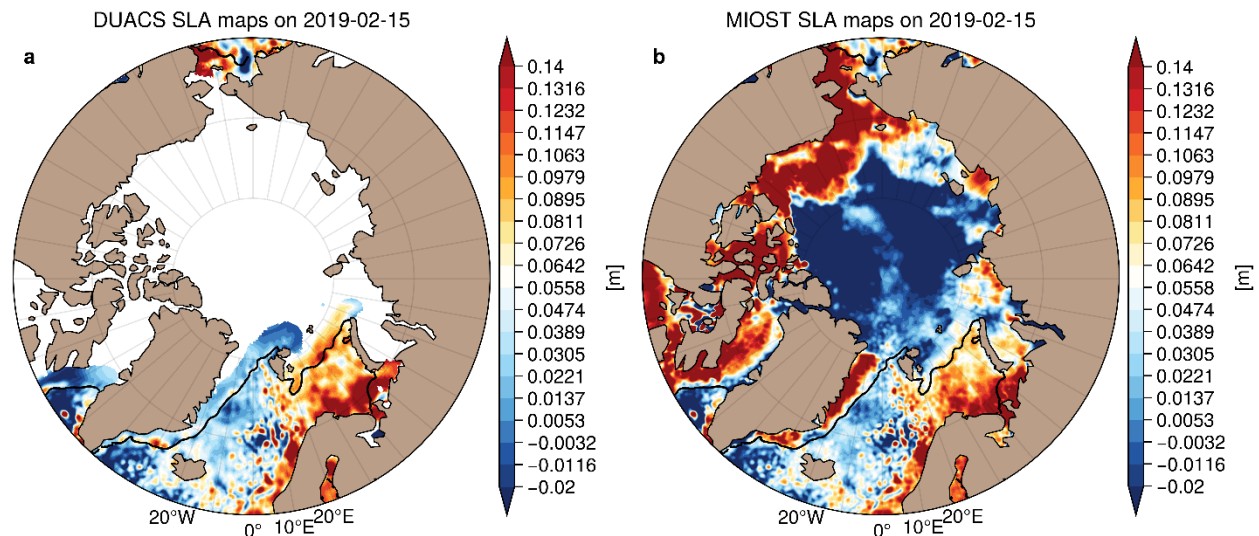

**Figure 5: Example of sea level anomaly maps on 2019-02-15 over the Arctic region constructed with the DUACS mapping approach a) and with the MIOST mapping approach b). The black line contour indicates the 15% sea-ice concentration from the OSI-SAF product**

From a global perspective, the MIOST maps are slightly more energetic than the DUACS maps as illustrated in Figure 6 with the variance maps and their differences. The difference between MIOST and DUACS variance maps (Figure 6c) indicates regions of higher variability in the MIOST maps (>10%) than in the DUACS maps, such as in the equatorial band, regions of low variability at mid-latitudes, coastal and polar regions. Tropical ocean regions are prone to lower SSH variability (10%) in the MIOST maps than in the DUACS maps.

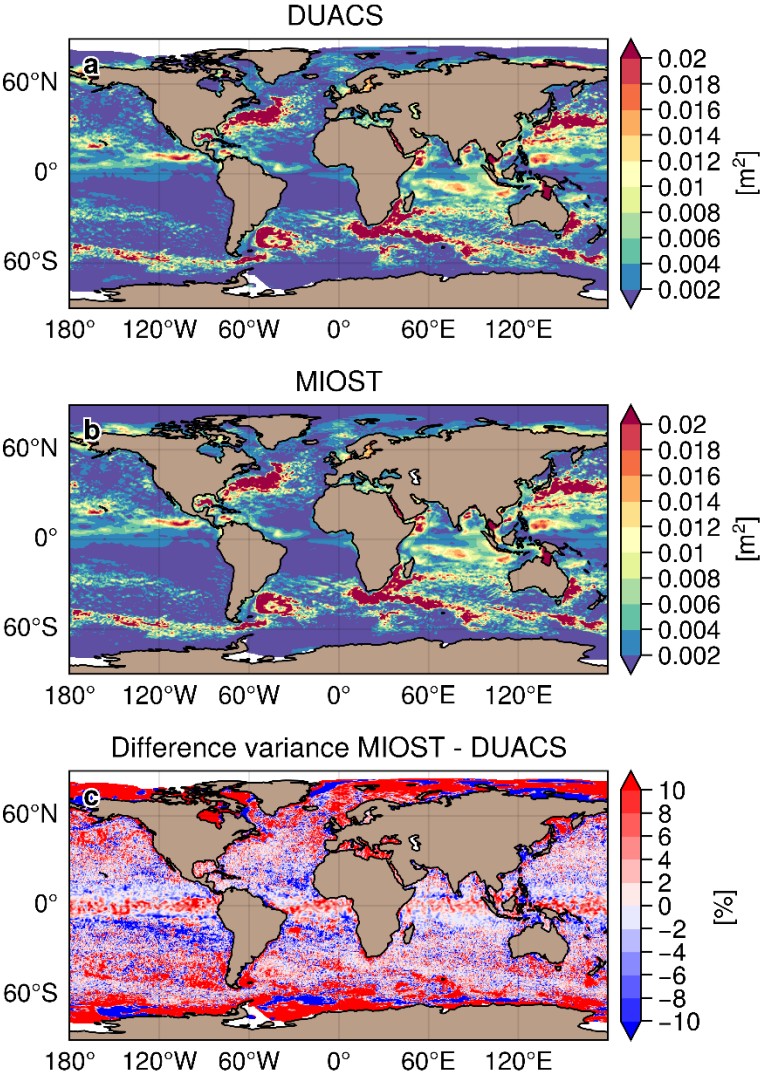

**Figure 6: Variance (in m²) of sea level anomaly maps constructed with a) the DUACS approach, b) the MIOST approach and b) difference between the MIOST and DUACS variance maps expressed in %**

The large SSH variability in the equatorial band of the MIOST maps is mainly associated with the equatorial wave components. The zonal wavenumber–frequency spectrum of SSH in the Pacific has been investigated in several study (e.g., Shinoda et al., 2009; Farrar, 2008, 2011) to examine the SSH variability associated with Tropical and Equatorial waves. Figure 7 shows contours of the base 10 logarithm of power in the wavenumber-frequency space calculated from SSH in the equatorial Pacific (region [180°E-280°E] [10°S:10°N]) for the period 2008 to 2018, for a) DUACS, b) MIOST with equatorial wave modes and c) in the GLORYS12V1 reanalysis (Lellouche et al., 2018). The rapid equatorial wave dynamics

are resolved in the GLORYS12v1 ocean numerical simulation (Figure 7c): the zonal wavenumber-frequency spectrum of the SSH in the Pacific reveals significant spectral peaks at periods close to 4 days, 5 days, and 7 days for a wavelength > 20° in longitude. These peaks are associated with inertia-gravity (Poincaré) waves. These SSH variabilities for time scales smaller than 10 days are filtered in the DUACS mapping approach (Figure 7a). In contrast, the MIOST multiscale mapping approach (*MIOST allsat-1 80% drifters + equatorial waves + L3 arctic*) resolves spectral peaks near 4 days, 5 days, and 7 days for wavelengths > 20° in longitude (Figure 7b). We show in the next section that these equatorial wave modes in MIOST also contributes to significantly reduce the mapping error in this region. For time scale > 10 days, each dataset has relatively similar spectral contents, particularly the energetic westward propagation of equatorial Rossby waves for negative wavenumbers.

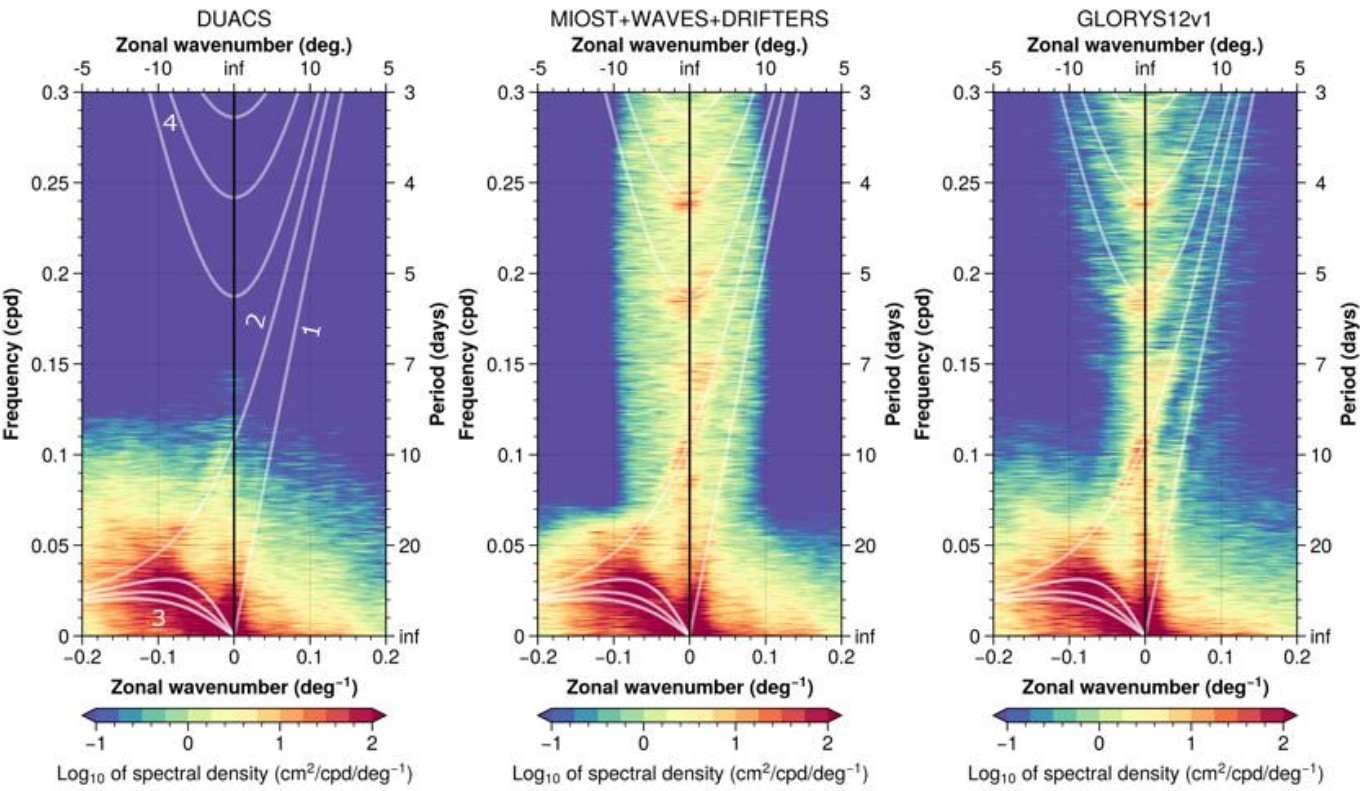

**Figure 7: Zonal wavenumber–frequency spectrum of SLA in the Equatorial Pacific computed for a) DUACS, b) MIOST with equatorial wave modes and c) in the GLORYS12V1 reanalysis. White lines represent the theoretical dispersion relation curves for equatorial waves corresponding to the Kelvin [1], Yanai [2], Rossby [3]and Poincaré [4] waves.**

## 4.2 Quantitative assessment

### 4.2.1 Mesoscale mapping assessments

The first assessment is a comparison of the *DUACS allsat-1* and *MIOST allsat-1* experiments. Both experiments aim to map the mesoscale circulation from altimetry data only. The SARAL/Altika altimeter and drifter sensors are not included in the mapping but are used as independent validation.

*Sea level anomaly quality*

The largest SSH mapping error $\sigma_{err}$ in *DUACS allsat-1* reaches 50-100 cm$^2$ in the western boundary surface current and over the continental plateaus (Figures 8a and 8b). In the offshore low variability region, the error variance is < 10 cm$^2$. Figures 8c and 8d show the difference in mapping error between the *MIOST allsat-1* and *DUACS allsat-1* experiments for all spatial and spatial scales between 65 and 500 km, respectively. Blue (red) pattern means a reduction (increase) of the mapping error in MIOST compared to DUACS. For all spatial scale considered, MIOST mapping errors are smaller than

those of DUACS, especially at mid-latitude with an average reduction in mapping error between 5% and 10%. The largest reduction in mapping error (~10%) is found in regions of high variability. In the inter-tropical region, MIOST and DUACS have similar scores. For spatial scale between 65 and 500 km, MIOST mapping errors are reduced by ~10% compared to DUACS in high variability region at mid-latitude. In low variability regions, the mapping error is between 3 and 4% smaller with MIOST than with DUACS, but the mapping errors are locally larger with MIOST than with DUACS: for example, in

the Argentine Sea, in the Siberian plateau and New Zealand plateau. Table 4 summarises the results of the comparison over different regions of interest (arctic, Antarctic, equatorial band, low variability region, and high variability region). Overall, the geostrophic flows in the MIOST SSH maps are closer to the independent SARAL/AltiKa observations than those in DUACS maps.

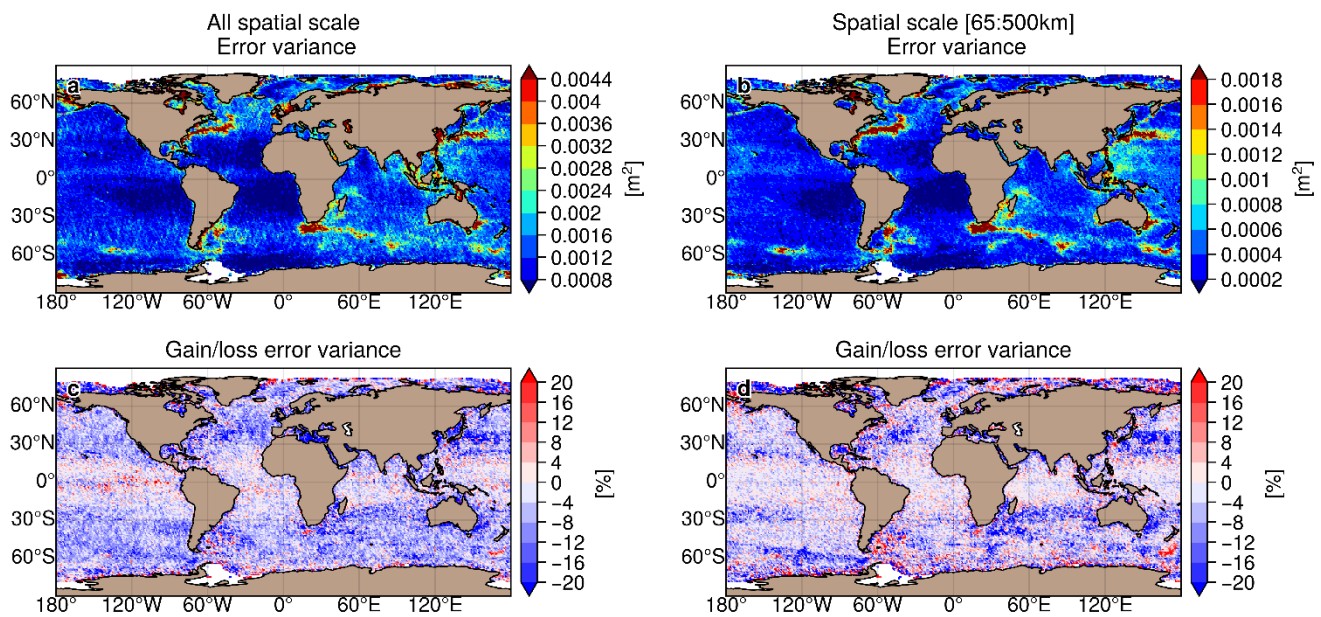

**Comparison SSH maps with independent SSH along-track**

**Figure 8: Variance of the difference SSH$_{map}$-SSH$_{alongtrack}$ computed for the DUACS allsat-1 experiment and in considering a) all spatial scale and b) spatial scale between 65 km and 500 km. Gain/loss of the mapping error variance of SLA in MIOST allsat-1 experiment relatively to the DUACS allsat-1 mapping error variance for c) all spatial scale and d) scale between 65 km and 500 km. Blue colour means a reduction of error variance in MIOST.**

**Table 4: Regionally averaged mapping error variance and gain/reduction of error variance on the SSH variable between MIOST and DUACS**

| Region | All spatial scale | | | Spatial scale [65:500km] | | |
|---|---|---|---|---|---|---|
| | Error variance DUACS [cm²] | Error variance MIOST [cm²] | Gain/loss error variance MIOST vs DUACS [%] | Error variance DUACS [cm²] | Error variance MIOST [cm²] | Gain/loss error variance MIOST vs DUACS [%] |
| Arctic | 23,18 | 23,17 | **-0,02** | 7.07 | 6.84 | **-3.31** |
| Antarctic | 33,07 | 31,13 | **-5,86** | 7.86 | 7.65 | **-2.64** |
| Equatorial band | 14,07 | 13,96 | **-0,80** | 4.66 | 4.67 | **+0.32** |
| Low variability - offshore | 12,54 | 11,81 | **-5,83** | 3.70 | 3.55 | **-4.11** |
| High variability - offshore | 30,87 | 27,71 | **-10,22** | 14.28 | 12.87 | **-9.86** |

*Geostrophic current quality*

Figures 9a and 9b show the validation against the independent drifter velocity data in terms of mapping error $\sigma_{err}$ for the zonal and meridional velocities. The largest mapping error $\sigma_{err}$ in DUACS reaches 300 to 400 cm².s⁻² in the western boundary surface current (e.g., the Gulfstream, the Kuroshio, Mozambique, and Agulhas currents). In offshore low variability region, the error variance is < 80 cm².s⁻². The differences in mapping error between MIOST and DUACS are

shown in Figures 9c and 9d for zonal and meridional velocities, respectively. Mapping errors are smaller in MIOST than in DUACS mainly in the core of the ocean gyres. In the intertropical region, the DUACS maps appear to be closer to the independent drifter velocities than MIOST. Table 5 summarises the results of the comparison over different regions of interest (arctic, Antarctic, equatorial band, low variability region, and high variability region). Overall, MIOST surface velocities are slightly closer to drifter velocities than the DUACS surface velocities.

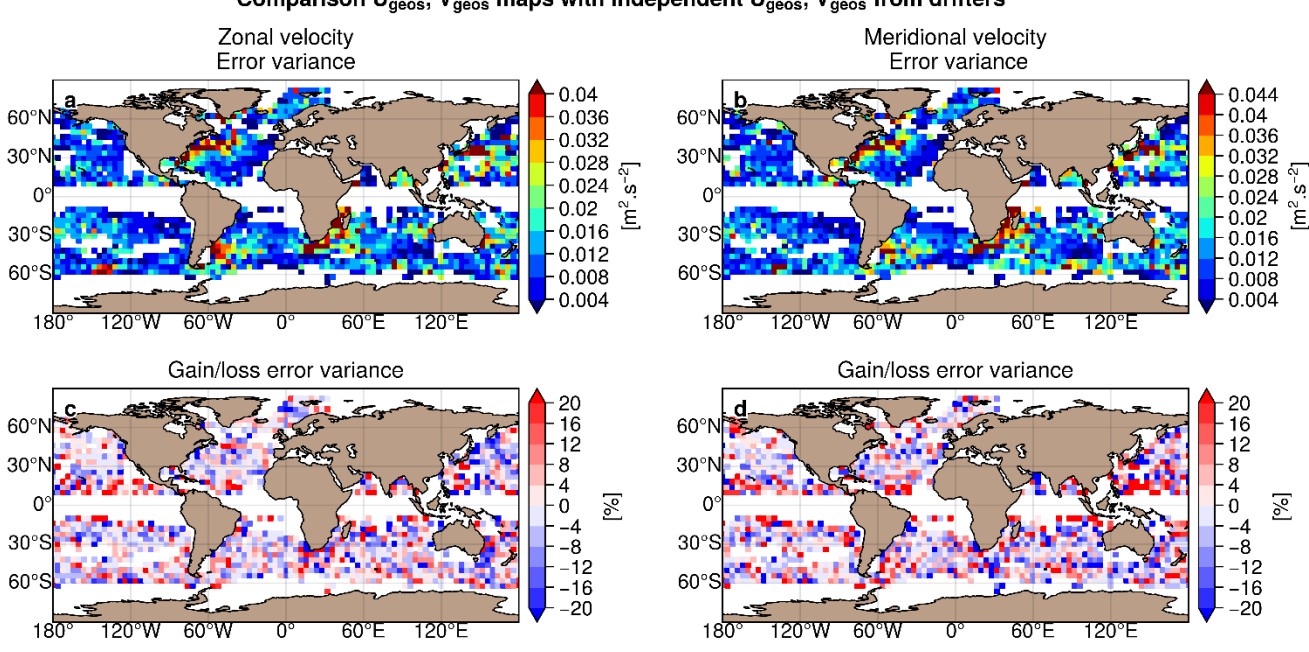

**Figure 9: Variance of the difference U$_{map}$-U$_{drifter}$ computed for the DUACS allsat-1 experiment and in considering a) the zonal velocity component and b) the meridional velocity component. Gain/loss of the mapping error variance of currents in MIOST allsat-1 experiment relatively to the DUACS allsat-1 mapping error variance for c) the zonal velocity component and d) the meridional velocity component. Blue colour means a reduction of error variance in MIOST.**

**Table 5: Regionally averaged mapping error variance and gain/reduction of error variance on the surface currents between MIOST and DUACS**

| Region | Zonal velocity | | | Meridional velocity | | |
|---|---|---|---|---|---|---|
| | Error variance DUACS [cm$^2$.s$^{-2}$] | Error variance MIOST [cm$^2$.s$^{-2}$] | Gain/loss error variance MIOST vs DUACS [%] | Error variance DUACS [cm$^2$.s$^{-2}$] | Error variance MIOST [cm$^2$.s$^{-2}$] | Gain/loss error variance MIOST vs DUACS [%] |
| Arctic | 153,17 | 148,78 | **-2,87** | 133,50 | 131,34 | **-1,62** |
| Antarctic | - | - | - | - | - | - |
| Equatorial band | - | - | - | - | - | - |
| Low variability - offshore | 130,36 | 128,52 | **-1,42** | 124,36 | 123,20 | **-0,94** |
| High variability - offshore | 385,86 | 372,40 | **-3,49** | 409,75 | 403,54 | **-1,51** |

### 4.2.2 Contribution of equatorial waves modes, and drifters' observations

The comparison of experiment *MIOST allsat-1 80% drifters + equatorial waves+ L3 arctic* with *MIOST allsat-1* examines the impact of the equatorial waves' mode and the drifters' observations in the MIOST mapping approach.

*Sea level anomaly quality*

The difference in mapping error between *MIOST allsat-1 80% drifters + equatorial waves+ L3 arctic* and *MIOST allsat-1* are shown in Figures 10a and 10b for all spatial and spatial scales between 65 and 500 km, respectively. For all spatial scale considered, we observe that the equatorial waves modes locally reduce the mapping error in the equatorial band by more than 10%. However, coastal equatorial regions (e.g., Indonesian Archipelago, western and Eastern part of Africa and South America) are prone to deterioration. This suggest that the equatorial wave mapping is not adapted in these coastal regions where different ocean processes are at play. In extra-equatorial regions, we evaluate the impact of drifter observations in MIOST. This impact is moderate on the SLA mapping (a few % of difference in the mapping error variance), with a reduction of error variance mainly in the high variability regions. For spatial scale between 65 and 500 km (Figure 10b), the equatorial waves modes deteriorate the mapping solution in the western and central Equatorial Pacific Ocean, in the Indian Ocean, while a reduced mapping error is found in the eastern Equatorial Pacific and the Equatorial Atlantic. In the extra-equatorial region, the impact of drifter observations remains moderate (with 1.5% error variance reduction in the high variability region). Overall, the drifters reduce the mapping errors primarily in regions of intense dynamics where the temporal sampling must be sufficiently accurate to properly map the rapid mesoscale dynamics. Table 6 summarises the results of the comparison over different regions of interest (arctic, Antarctic, equatorial band, low variability region, and high variability region).

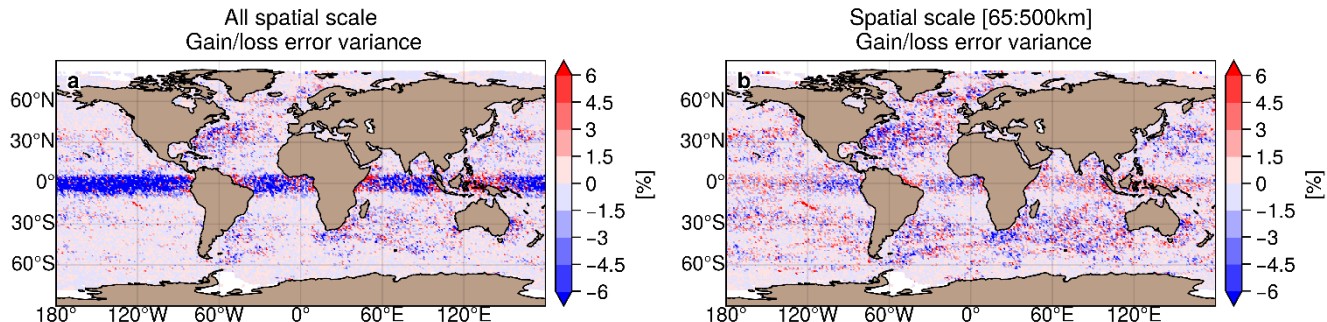

**Figure 10: Gain/loss of the mapping error variance of SLA in MIOST allsat-1 80% drifters + equatorial waves+ L3 arctic experiment relatively to the MIOST allsat-1 mapping error variance for a) all spatial scale and b) scale between 65 km and 500 km. Blue colour means a reduction of error variance in MIOST when drifters are included in the mapping and with equatorial waves parametrization**

**Table 6: Regionally averaged mapping error variance and gain/reduction of error variance on the SSH variable between *MIOST allsat-1 80% drifters + equatorial waves+ L3 arctic* and *MIOST allsat-1***

| Region | All spatial scale | | | Spatial scale [65:500 km] | | |
|---|---|---|---|---|---|---|
| | Error variance MIOST allsat-1 [cm$^2$] | Error variance MIOST allsat-1 80% drifters + equatorial waves+ L3 arctic [cm$^2$] | Gain/loss error variance MIOST allsat-1 80% drifters + equatorial waves+ L3 arctic vs MIOST allsat-1 [%] | Error variance MIOST allsat-1 80% drifters + equatorial waves+ L3 arctic [cm$^2$] | Error variance MIOST allsat-1 [cm$^2$] | Gain/loss error variance MIOST allsat-1 80% drifters + equatorial waves+ L3 arctic vs MIOST allsat-1 [%] |
| Arctic | 23,17 | 23,18 | **+0,02** | 6,84 | 6,84 | **+0,00** |
| Antarctic | 31,13 | 31,14 | **+0,02** | 7,65 | 7,65 | **+0,01** |
| Equatorial band | 13,96 | 13,53 | **-3,03** | 4,67 | 4,69 | **+0,32** |
| Low variability - offshore | 11,81 | 11,72 | **-0,77** | 3,55 | 3,54 | **-0,10** |
| High variability - offshore | 27,71 | 27,42 | **-1,06** | 12,87 | 12,67 | **-1,54** |

*Geostrophic current quality*

The difference in mapping error of surface geostrophic currents between *MIOST allsat-1 80% drifters + equatorial waves+ L3 arctic* and *MIOST allsat-1* are shown in Figures 11a and 11b for the zonal component and the meridional component of the velocity, respectively. It is difficult to draw conclusions from this diagnosis: the mapping errors are reduced with MIOST in some regions in the tropics (such as the Bay of Bengal), in the Kuroshio extension. Overall, the contribution of drifters remains moderate for the restitution of geostrophic currents (only a few % improvement in the open ocean) as summarized in Table 7.

**Comparison U$_{geos}$, V$_{geos}$ maps with independent U$_{geos}$, V$_{geos}$ from drifters**

**Figure 11: Gain/loss of the mapping error variance of currents in MIOST allsat-1 80% drifters + equatorial waves+L3 arctic experiment relatively to the MIOST allsat-1 mapping error variance for c) the zonal velocity component and d) the meridional velocity component. Blue colour means a reduction of error in MIOST when drifters are included in the mapping and with equatorial waves parametrization**

**Table 7: Regionally averaged mapping error variance and gain/reduction of error variance on the surface currents between**
*MIOST allsat-1* **and** *MIOST allsat-1 80% drifters + equatorial waves+ L3 arctic*

| Region | Zonal velocity | | | Meridional velocity | | |
|---|---|---|---|---|---|---|
| | Error variance MIOST allsat-1 [cm$^2$.s$^{-2}$] | Error variance MIOST allsat-1 80% drifters + equatorial waves+ L3 arctic [cm$^2$.s$^{-2}$] | Gain/loss error variance MIOST allsat-1 80% drifters + equatorial waves+ L3 arctic vs MIOST allsat-1 [%] | Error variance MIOST allsat-1 [cm$^2$.s$^{-2}$] | Error variance MIOST allsat-1 80% drifters + equatorial waves+ L3 arctic [cm$^2$.s$^{-2}$] | Gain/loss error variance MIOST allsat-1 80% drifters + equatorial waves+ L3 arctics vs MIOST allsat-1 [%] |
| Arctic | 148,78 | 145,04 | **-2,51** | 131,34 | 127,83 | **-2,67** |
| Antarctic | - | - | - | - | - | - |
| Equatorial band | - | - | - | - | - | - |
| Low variability - offshore | 128,52 | 127,80 | **-0,56** | 123,20 | 122,04 | **-0,94** |
| High variability - offshore | 372,40 | 366,81 | **-1,50** | 403,54 | 400,90 | **-0,65** |

### 4.2.3 Overall assessment

The comparison of the *MIOST allsat-1 80% drifters + equatorial waves+L3 arctic* and *DUACS allsat-1* experiments allows to evaluate the complete MIOST product distributed to users against the DUACS method.

*Sea level anomaly quality*

The difference in mapping error between *MIOST allsat-1 80% drifters + equatorial waves+ L3 arctic* and *DUACS allsat-1* are shown in Figures 12a and 12b for all spatial and spatial scales between 65 and 500 km, respectively. We have the same pattern as found in the previous sections: for all spatial scale considered (Figure 12a), the equatorial waves modes help to reduce the mapping error variance in the equatorial band by more than 20% locally. At mid-latitude, the mapping error are between 5% and 10% smaller with MIOST than with DUACS. For spatial scales between 65 and 500 km, MIOST and DUACS solutions are globally equivalent, except in the high variability region where the mapping error is between 10% and 20% smaller with MIOST than with DUACS. The mapping errors are locally larger with MIOST than with DUACS in regions where the circulation interact with bathymetry feature such as in the Argentine Sea, near the Siberian plateau and New Zealand plateau. Table 8 summarises the results of the comparison over different regions of interest: mapping errors are ~11% smaller in high variability region in MIOST than in DUACS. In other regions, the errors are ~3-6% smaller.

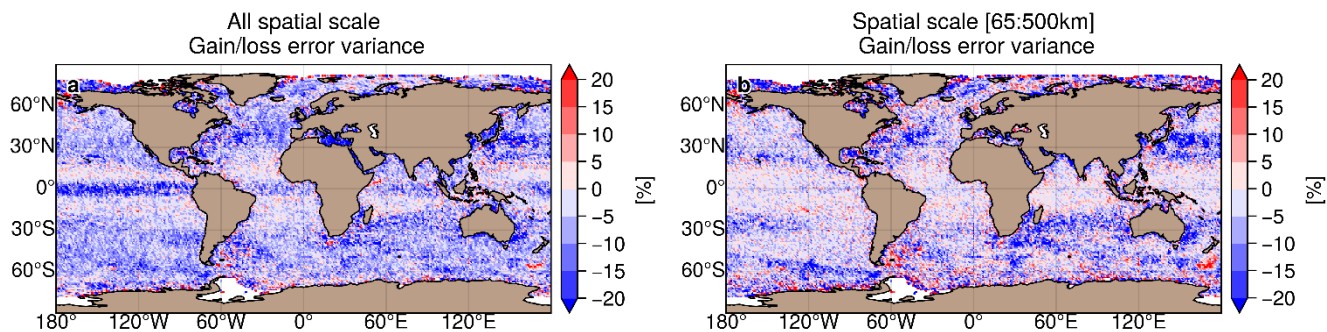

**Figure 12: Gain/loss of the mapping error variance of SLA in MIOST allsat-1 80% drifters + equatorial waves+ L3 arctic experiment relatively to the DUACS allsat-1 mapping error variance for a) all spatial scale and b) scale between 65 km and 500 km. Blue colour means a reduction of error variance in MIOST.**

**Table 8: Regionally averaged mapping error variance and gain/reduction of error variance on the SSH variable between *MIOST allsat-1 80% drifters + equatorial waves+ L3 arctic* and *DUACS allsat-1***

| Region | All spatial scale | | | Spatial scale [65:500 km] | | |
|---|---|---|---|---|---|---|
| | Error variance DUACS allsat-1 [cm²] | Error variance MIOST allsat-1 80% drifters + equatorial waves+ L3 arctic [cm²] | Gain/loss error variance MIOST allsat-1 80% drifters + equatorial waves+ L3 arctic vs DUACS allsat-1 [%] | Error variance DUACS allsat-1 [cm²] | Error variance MIOST allsat-1 80% drifters + equatorial waves+ L3 arctic [cm²] | Gain/loss error variance MIOST allsat-1 80% drifters + equatorial waves+ L3 arctic vs DUACS allsat-1 [%] |
| Arctic | 23,18 | 23,18 | **+0,01** | 7,07 | 6,84 | **-3,31** |
| Antarctic | 33,07 | 31,14 | **-5,85** | 7,86 | 7,65 | **-2,63** |
| Equatorial band | 14,07 | 13,53 | **-3,81** | 4,66 | 4,69 | **+0,64** |
| Low variability - offshore | 12,54 | 11,72 | **-6,56** | 3,70 | 3,54 | **-4,20** |
| High variability - offshore | 30,87 | 27,42 | **-11,16** | 14,28 | 12,67 | **-11,24** |

ss

*Geostrophic current quality*

The difference in mapping error of surface geostrophic currents between *MIOST allsat-1 80% drifters + equatorial waves+ L3 arctic* and *DUACS allsat-1* are shown in Figures 13a and 13b for the zonal component and the meridional component of the velocity, respectively. The mapping errors are globally smaller in MIOST than in DUACS, particularly in the high variability regions. In the tropical regions, DUACS outperforms MIOST for reconstructing the surface geostrophic velocities. Overall, the mapping errors are on average between ~2% and 5% smaller with MIOST than with DUACS (Table 9).

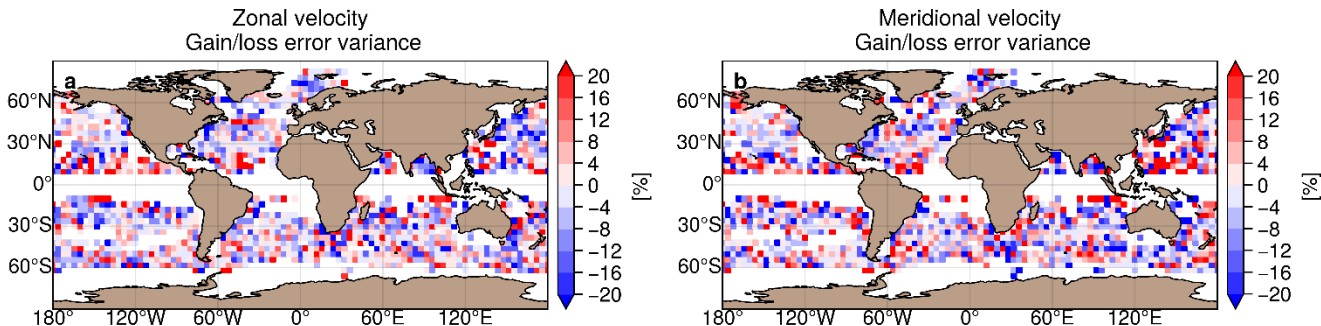

**Figure 13: Gain/loss of the mapping error variance of currents in MIOST allsat-1 80% drifters + equatorial waves + L3 arctic experiment relatively to the DUACS allsat-1 mapping error variance for c) the zonal velocity component and d) the meridional velocity component. Blue colour means a reduction of error in MIOST**

**Table 9: Regionally averaged mapping error variance and gain/reduction of error variance on the surface currents between** *MIOST allsat-1 80% drifters + equatorial waves+ L3 arctic* **and** *DUACS allsat-1*

| Region | Zonal velocity | | | Meridional velocity | | |
|---|---|---|---|---|---|---|
| | Error variance DUACS allsat-1 [cm$^2$.s$^{-2}$] | Error variance MIOST allsat-1 80% drifters + equatorial waves+ L3 arctic [cm$^2$.s$^{-2}$] | Gain/loss error variance MIOST allsat-1 80% drifters + equatorial waves+ L3 arctic vs DUACS allsat-1 [%] | Error variance DUACS allsat-1 [cm$^2$.s$^{-2}$] | Error variance MIOST allsat-1 80% drifters + equatorial waves+ L3 arctic [cm$^2$.s$^{-2}$] | Gain/loss error variance MIOST allsat-1 80% drifters + equatorial waves+ L3 arctic vs DUACS allsat-1 [%] |
| Arctic | 153,17 | 145,04 | **-5.31** | 133,50 | 127,83 | **-4.25** |
| Antarctic | - | - | - | - | - | - |
| Equatorial band | - | - | - | - | - | - |
| Low variability - offshore | 130,36 | 127,80 | **-1.96** | 124,36 | 122,04 | **-1.87** |
| High variability - offshore | 385,86 | 366,81 | **-4.94** | 409,75 | 400,90 | **-2.16** |

*Effective resolution*

The effective spatial resolution quantifies the minimum spatial scale resolved in the maps (Ballarotta et al., 2019). Maps of the effective spatial resolution (expressed in kilometres) are presented in Figure 14a and Figure 14b for *DUACS allsat-1* and *MIOST allsat-1 80% drifters + equatorial waves+ L3 arctic*, respectively. For each experiment, the effective spatial resolution varies from ~500 km at the equator to ~100 km at high altitude, and a mean value at mid-latitude close to 200 km. The difference in effective spatial resolution between the two experiments is shown in Figure 14c. The resolution of the SLA maps of the MIOST experiment is overall finer than in the SLA maps of the DUACS experiment. It is between 5% and 10% finer than the DUACS maps in regions of high variability (Gulfstream, Kuroshio, and Agulhas regions), in the Atlantic and equatorial Pacific, and in the Norwegian and Greenland seas. Some regions (e.g., tropical regions, coastal regions, the East

China Sea, the New Zealand Shelf, or the Argentine Sea) are subject to a coarser effective resolution in MIOST maps than in DUACS maps. These regions will require further investigation in the near future.

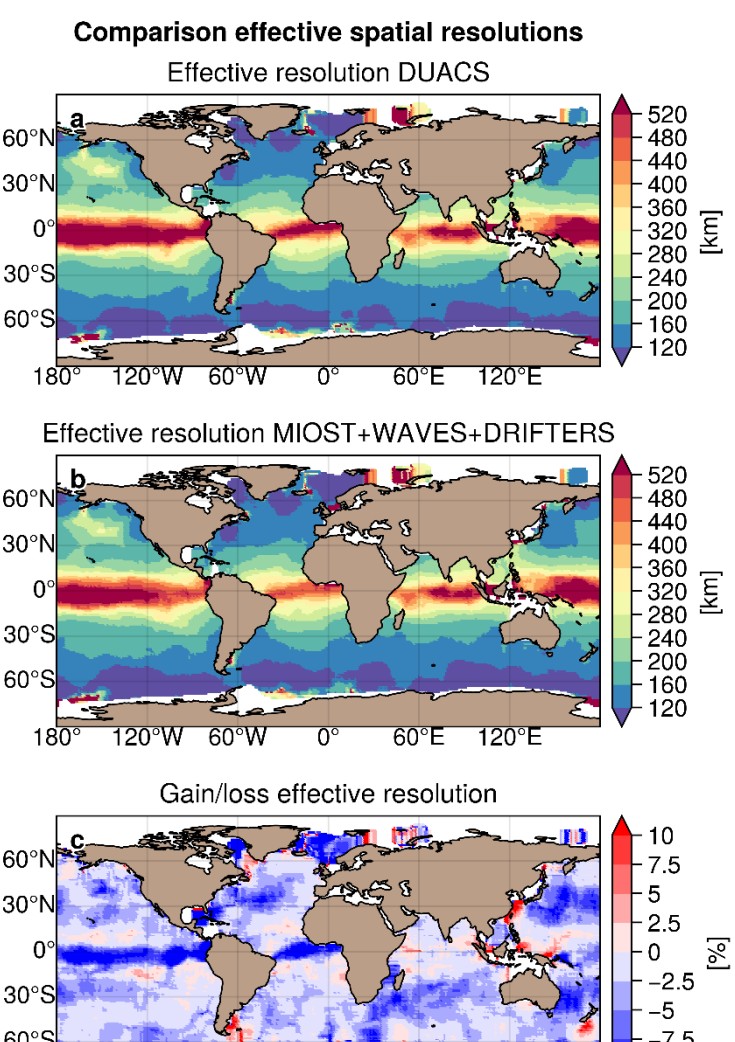

**Figure 14: Maps of effective spatial resolution (in km) for a) the DUACS allsat-1 and b) MIOST allsat-1 80% drifters + equatorial waves+L3 arctic experiments; and c) gain/loss of effective resolution (in %) between MIOST and DUACS. Blue means finer resolution in MIOST than in DUACS**

## 5 Summary & Conclusions

Ubelmann et al (2021, 2022) evaluated the multiscale & multivariate mapping approach in Observing System Simulation Experiment (OSSE) and Observing System Experiment (OSE) for simultaneous mapping of mesoscale circulation, coherent

internal tides, surface geostrophic and ageostrophic velocities. Here, we extend the application of the MIOST solution to the

455 simultaneous mapping of equatorial waves and mesoscale circulation from real observations. Furthermore, we investigate the levels of mapping improvement by enhancing the sampling of the ocean surface state with in-situ data and altimetry data in the Arctic sea-ice regions. We found that the Arctic leads SSH observations allow to significantly improve the monitoring coverage in this remote region. The gap-free maps, proposed with MIOST, hence offer to the end-users the opportunity to study the arctic surface circulation and its connections to the subpolar and mid-latitude regions. It is important to mention

that this polar mapping will need to be validated against independent data in the near future. Drifters' observations have a moderate impact in the mapping. They mainly contribute to reduce mapping errors in regions of intense dynamics where the temporal sampling must be accurate enough to properly map the rapid mesoscale dynamics. It is important note that drifter observations can potentially improve surface circulation in areas not or poorly sampled by altimeters. Therefore, their impact on the sea level reconstruction may be larger over period of weak altimeter sampling.

The ocean surface circulation involves a superposition of processes acting at widely different spatial and temporal scales, from the geostrophic large-scale and slow varying flow to the mesoscale turbulent eddies and at even smaller scale, the mixing generated by the internal wave field. It is also important to mention that the DUACS maps are constructed from altimetry data using an interpolation method optimized for mapping mesoscale variability. Consequently, some ocean

surface variabilities are not or poorly represented in these DUACS maps: equatorial wave dynamics is thus part of the filtered ocean signals in DUACS. The multiscale approach allows to decompose the observed SSH into various physical contributions. Here, we explored and validated the possibility of improving the content of altimetry maps by simultaneously estimating the ocean mesoscale circulations as well as the equatorial wave dynamics associated to the Tropical Instability Waves and Poincaré waves. We show that mapping these ocean surface variabilities from altimeter observations broadens

the spectrum of mappable space-time scales and reduces mapping errors by almost 20% locally relative to independent data, primarily in the equatorial Pacific and Atlantic basins. This is possible because the spatio-temporal coverage of the altimeter data allows to sample large scale waves of 4-day periods and longer. At global scale, we also found that, compared to the operational DUACS mapping approach, MIOST approach improves the surface mesoscale circulations in regions of high variability. Consequently, the effective resolution of the maps produced by the multiscale approach is finer than the DUACS

maps, particularly in the western boundary currents and in the equatorial band.

This experimental product is currently available on the AVISO+ (Archivage, Validation et Interprétation des données des Satellites Océanographiques) website (see the Data Availability section for more detail), but our results suggest that the multiscale & multivariate mapping approach is very promising for use in an operational context. It is also worth mentioning

that several other global gridded products exist as an alternative to the DUACS/MIOST products which provide only the geostrophic part of the surface current. Examples of these other products that provide a broader spectrum of ocean surface current variability (e.g., the total surface currents) include, 1) the Copernicus GLORYS12v1 global ocean reanalysis

(Lellouche et al., 2018; https://doi.org/10.48670/moi-00021), 2) the Copernicus GLOBCURRENT product (Rio et al., 2014; https://doi.org/10.48670/moi-00050), or 3) the OSCAR product (Dohan, 2021; https://doi.org/10.5067/OSCAR-25I20) distributed by the NASA-JPL Distributed Physical Oceanography Active Archive Center (PO. DAAC).

To conclude, these results pave the way for the exploration of new types of ocean signals that may eventually be mapped with MIOST from remote sensing and in situ observations. Future work could consist of enriching the MIOST components in considering oceanic signals missing in the maps and yet captured by observing systems: for example, in mapping high frequency signals such as the near-inertial oscillation from drifter observations, in using SSH leads products in the Southern Ocean (Auger et al., 2022); or by enhancing the SLA maps content with dynamical model approach (Ubelmann et al, 2015) or Artificial Intelligence methods (Beauchamp et al., 2020).

## 6 Data Availability

The MIOST gridded products (https://doi.org/10.24400/527896/a01-2022.009, Ballarotta et al., 2022) are hosted on the AVISO+ (Archivage, Validation et Interprétation des données des Satellites Océanographiques) website at the following repository: https://data.aviso.altimetry.fr/aviso-gateway/data/SLA_MIOST_alti_drifters/.
The reference DUACS maps are hosted on the E.U. Copernicus Marine Service portal (https://doi.org/10.48670/moi-00146).

## 7 Product description

The multiscale & multivariate products are distributed on a regular grid: the spatial grid extends from 0°E to 360°E in longitude, 80°S to 90°N in latitude, with a grid spacing of 0.1°; the temporal grid covers the period 2016-07-01 to 2020-06-30 with a time step of 1 day. The dataset is distributed in netCDF4 format. Each netCDF file contains 6 variables: sla, adt, ugosa, vgosa, ugos, and vgos.

## 8 Acknowledgements

The work presented here was carried out in the framework of the DUACS-R&D project funded by CNES. The authors would like to thank the AVISO+ (Archivage, Validation et Interprétation des données des Satellites Océanographiques) team for their support and expertise in the distribution of the dataset. We are grateful to the three anonymous reviewers for their comments and suggestions to improve the manuscript.

```
dimensions:
        time = 1 ;
        latitude = 1702 ;
        longitude = 3600 ;
        bounds = 2 ;
variables:
        int sla(time, latitude, longitude) ;
                sla:_FillValue = -2147483647 ;
                sla:coordinates = "longitude latitude" ;
                sla:grid_mapping = "crs" ;
                sla:long_name = "Sea level anomaly" ;
                sla:standard_name = "sea_surface_height_above_sea_level" ;
                sla:units = "m" ;
                sla:scale_factor = 0.0001 ;
        int ugosa(time, latitude, longitude) ;
                ugosa:_FillValue = -2147483647 ;
                ugosa:coordinates = "longitude latitude" ;
                ugosa:grid_mapping = "crs" ;
                ugosa:long_name = "Geostrophic velocity anomalies: zonal component" ;
                ugosa:standard_name = "surface_geostrophic_eastward_sea_water_velocity_assuming_sea_level_for_geoid" ;
                ugosa:units = "m" ;
                ugosa:scale_factor = 0.0001 ;
        int vgosa(time, latitude, longitude) ;
                vgosa:_FillValue = -2147483647 ;
                vgosa:coordinates = "longitude latitude" ;
                vgosa:grid_mapping = "crs" ;
                vgosa:long_name = "Geostrophic velocity anomalies: meridional component" ;
                vgosa:standard_name = "surface_geostrophic_northward_sea_water_velocity_assuming_sea_level_for_geoid" ;
                vgosa:units = "m" ;
                vgosa:scale_factor = 0.0001 ;
        int adt(time, latitude, longitude) ;
                adt:_FillValue = -2147483647 ;
                adt:coordinates = "longitude latitude" ;
                adt:grid_mapping = "crs" ;
                adt:long_name = "Absolute dynamic topography" ;
                adt:standard_name = "sea_surface_height_above_sea_level" ;
                adt:units = "m" ;
                adt:scale_factor = 0.0001 ;
        int ugos(time, latitude, longitude) ;
                ugos:_FillValue = -2147483647 ;
                ugos:coordinates = "longitude latitude" ;
                ugos:grid_mapping = "crs" ;
                ugos:long_name = "Absolute geostrophic velocity: zonal component" ;
                ugos:standard_name = "surface_geostrophic_eastward_sea_water_velocity" ;
                ugos:units = "m" ;
                ugos:scale_factor = 0.0001 ;
        int vgos(time, latitude, longitude) ;
                vgos:_FillValue = -2147483647 ;
                vgos:coordinates = "longitude latitude" ;
                vgos:grid_mapping = "crs" ;
                vgos:long_name = "Absolute geostrophic velocity: meridional component" ;
                vgos:standard_name = "surface_geostrophic_northward_sea_water_velocity" ;
                vgos:units = "m" ;
                vgos:scale_factor = 0.0001 ;
        float latitude(latitude) ;
                latitude:_FillValue = NaNf ;
                latitude:axis = "Y" ;
                latitude:long_name = "Latitude" ;
                latitude:standard_name = "latitude" ;
                latitude:units = "degrees_north" ;
                latitude:valid_max = 90. ;
                latitude:valid_min = -80.1 ;
                latitude:bounds = "latitude_bounds" ;
        float longitude(longitude) ;
                longitude:_FillValue = NaNf ;
                longitude:axis = "X" ;
                longitude:long_name = "Longitude" ;
                longitude:standard_name = "longitude" ;
                longitude:units = "degrees_east" ;
                longitude:valid_max = 359.9 ;
                longitude:valid_min = 0. ;
                longitude:bounds = "longitude_bounds" ;
        double time(time) ;
                time:_FillValue = NaN ;
                time:units = "days since 1950-01-01 00:00:00" ;
                time:calendar = "gregorian" ;
                time:axis = "T" ;
                time:standard_name = "time" ;
        float longitude_bounds(longitude, bounds) ;
                longitude_bounds:_FillValue = NaNf ;
        float latitude_bounds(latitude, bounds) ;
                latitude_bounds:_FillValue = NaNf ;
```

**Appendix A**

**2.2.1 The Optimal Interpolation (DUACS mapping approach)**

The DUACS mapping approach constructs a SSH field on a regular grid by combining measurements from various altimeters. It is based on a global suboptimal space-time objective analysis that considers along-track correlated errors as described for instance in Ducet et al., (2000) or Le Traon et al. (2003). The mathematical formulation, knows as Optimal Interpolation, is described hereafter.

We assume a state to estimate, denoted x, and partial observations, denoted y, which can be related to the state by a linear

operator H such as:

$$y = Hx + \epsilon \tag{A1}$$

where $\epsilon$ is an independent signal (e.g., observation error) not related to the state. If we define B the covariance matrix of x and $R$ the covariance matrix of $\epsilon$, both variables being assumed Gaussian, then the linear estimate is written:

$$x^a = BH^T(HBH^T - R)^{-1}y \tag{A2}$$

The observation vector $y$ represents the SLA observations. The state vector x is the gridded SLA. The operator $H$ (formally a tri-linear interpolator transforming the gridded state SLA to the equivalent along-track SLA) is not considered explicitly. The matrices $BH^T$ and $HBH^T$, representing the covariance of the signal in the (grid, obs) and (obs, obs) spaces, are directly

written with the analytical formula of the Arhan and Colin de Verdière (1985) covariance model as described in Ducet et al., (2000), Le Traon et al. (2003) or Pujol et al. (2016):

$$C(x,y,t) = \left(1 + ar + \frac{1}{6}(ar)^2 - \frac{1}{6}(ar)^3\right)e^{-ar}e^{-(\frac{t}{L_t})^2} \tag{A3}$$

$$r = \sqrt{\left(\frac{x - C_{px}t}{L_x}\right)^2 + \left(\frac{y - C_{py}t}{L_y}\right)^2} \tag{A4}$$

where, x, y, t corresponds the zonal, meridional and temporal position, Lx, Ly, Lt are the zonal, meridional and temporal

decorrelation scale, $C_{px}$ and $C_{py}$ the phase speed, a is a constant (3.337).

This covariance model is mainly optimized for mesoscale signal reconstruction. The $R$ matrix represents the representativity and instrumental errors. Since the covariance of mesoscale SLA is assumed to vanish beyond a few hundreds of kilometres in space and beyond 10–20 days in time (Le Traon & Dibarboure, 2002), separate inversions are performed locally selecting observations over time and space windows adjusted to these values. In practice, since the number of observations is limited

to less than 1000 (Le Traon et al, 1998), the inversion in observation space is computationally manageable. More details on the map production are given in Pujol et al. (2016).

In DUACS, the geostrophic current $(U_g, V_g)$ is then directly derived from the mapped SSH:

$$U_g(x,y) = -\frac{g}{f_c}\frac{\partial SSH(x,y)}{\partial y} \tag{A5}$$

$$V_g(x,y) = \frac{g}{f_c}\frac{\partial SSH(x,y)}{\partial x} \tag{A6}$$

where $g$ is the gravity, $f_c$ is the Coriolis frequency, which is a function of latitude.

### 2.2.2 A multiscale & multivariate mapping approach

The Optimal Interpolation requires the inversion of a matrix of the same size as the observation vector y. When the number of observations exceeds the size of the state to resolve, it can be interesting to use an equivalent formulation given by the Sherman-Morrison-Woodbury transformation, allowing an inversion in state space, with a matrix of the size of the state

vector x,

$$x^a = (H^T \llbracket R^{-1}H + B^{-1})\rrbracket^{-1} H^T R^{-1} y \tag{A7}$$

The formulation of the multiscale & multivariate mapping algorithm is detailed in Ubelmann et al. (2022). We here recall the main principle. We consider an extended state vector x composed by N physical components that will be later assumed independent. In this study N=3 for 1) geostrophy and equatorial waves: 2) Tropical Instability Waves (TIW) and 3) Poincaré

waves:

$$x = (x_1^T, \ldots, x_N^T)^T \tag{A8}$$

Each component $x_k$ represents the state of the surface topography and surface current to be resolved in the grid space, noted

$x_k = (h_k^T, u_k^T, v_k^T)^T$. The key aspect of the method is a rank reduction of the state vector, through a subcomponent decomposition, such as $x_k$ can be written as:

$$x_k = \begin{bmatrix} \Gamma_{k,h} \\ \Gamma_{k,u} \\ \Gamma_{k,v} \end{bmatrix} \eta_k = \Gamma_k \eta_k \tag{A9}$$

where $\eta_k$ is the reduced state vector for component k, $\Gamma_{k,h}$, $\Gamma_{k,u}$, and $\Gamma_{k,v}$ are the subcomponent matrices expressed in

topography and currents, respectively. Note that for some components, one of the blocks can be set to zeros (e.g., if geostrophy component is considered with zero contribution on SSH, which is the case for the equatorial wave components). Their concatenation is called $\Gamma_k$ which is the matrix transforming the reduced state vector in the grid space for topography and currents. In practice, $\Gamma_k$ will be a wavelet decomposition of the time-space domain, with elements of appropriate temporal and spatial scales to represent the component k. These wavelet scales, and their specified variance set with a

diagonal matrix noted $Q_k$, will define the equivalent covariance model $B_k$ in the grid space for component k:

$$B_k = \Gamma_k Q_k \Gamma_k^T \tag{A10}$$

The observation vector y is also extended to the observed surface topography and surface current noted $y = (h^{oT}, u_r^{oT})^T$. Then, if $H_k$ is the observation operator for component k (from grid space to observation space), we note $G_k = H_k \Gamma_k$ the subcomponent matrix expressed in observation space. In these conditions, the observation vector y is the sum of all component contributions plus the unexplained signal $\epsilon$ (instrument error and representativity),

$$y = \sum_{k=1}^{N} G_k \eta_k + \epsilon \qquad (A11)$$

If we use the notation $\eta = (\eta_1^T, \dots, \eta_k^T)^T$ for the concatenation of the subcomponent state vectors, and $G = (G_1, \dots, G_N)$, then we have,

$$y = G\eta + \epsilon \qquad (A12)$$

Applying the same transformation from Equation A1, Equation A2, and Equation A7 to the reduced state vector η, the global solution is written:

$$\eta^a = (G^T R^{-1} G + Q^{-1})^{-1} G^T R^{-1} y \qquad (A13)$$

where $Q$ is the covariance matrix of η, expressed as the concatenation of the diagonal matrices $Q_k$ for each component. Finally, the solution in the reduced-space projects into the grid space with the following relation:

$$x_a = \Gamma \eta^a \qquad (A14)$$

In practice, to solve Equation A13, each block of $G$ is directly filled from the analytical expression of the reduced-space elements constituting the columns of the matrix. Also, in many situations, the $(G^T R^{-1} G + Q^{-1})$ matrix, noted A hereafter, would be too large to be inverted (as required by Equation A13 explicitly). We use a preconditioned conjugate gradient method to solve $\eta = A^{-1}z$ where $z = G^T R^{-1} y$ is computed initially from $G$ and the observation vector $y$. The algorithm involves many iterations of Aη computations for updated η until convergence is reached (when Aη approaches z). Note that if A is too large to be written explicitly, the result Aη can still be computed in two steps from a matrix multiplication of $G$ then of $G^T$. Once the solution η is obtained, the projection in physical grid space given by Equation A14 is applied sequentially, by summing the analytical expression of the ripples applied to grid coordinates (the columns of Γ), separately for each component k. As in any inversion based on linear analysis, the result strongly relies on the choice of covariance models, here defined by the reduced elements of each component.

*Geostrophy component*

Geostrophy is the component that has a signature on both topography and currents, and on which some synergy between altimetry and drifter observations can be expected. Following the formulation provided in Ubelmann et al. (2021), we define here the gridded variable $H_1$ to resolve, and the corresponding gridded geostrophic current field $(U_1, V_1)$ writes

$$\begin{cases} U_1 = -\dfrac{g}{f_c}\dfrac{\partial H_1}{\partial y} \\[2mm] V_1 = \dfrac{g}{f_c}\dfrac{\partial H_1}{dx} \end{cases} \qquad (A15)$$

The proposed reduced state for geostrophy is based on an element decomposition of $H_1$, expressed by $\Gamma_{1,h}$ with wavelets of various wavelength and temporal extensions. This will allow to approximate the standard covariance models used in altimetry mapping, accounting for specific variations with wavelength and time. A given p element of the decomposition $\Gamma_{1,h}$ is expressed as follows:

$$\Gamma_{1,h}[i,p] = \cos\big(k_{x,p}(x_i - x_p) + k_{y,p}(y_i - y_p) + \Phi_p \big) * f_{tap}\left(\frac{x_i - x_p}{L_{x_p}}, \frac{y_i - y_p}{L_{y_p}}, \frac{t_i - t_p}{L_{t_p}}\right) \qquad (A16)$$

where the ith line of the matrix stands for a given grid index of coordinates $(x_i, y_i, t_i)$. For the ensemble of p, $\Phi_p$ is alternatively 0 and $\pi/2$, such as all subcomponents are defined by pairs of sine and cosine functions to allow the phase degree of freedom. $k_{x,p}$ and $k_{y,p}$ are zonal and meridional wavenumbers respectively, set to vary in the mappable mesoscale range (between 80 km and 900 km with a spacing inversely proportional to the wavelet extensions, allowing to represent a signal of any intermediate wavelength). $(x_p, y_p, t_p)$ are the coordinates of a space-time pavement. The function $f_{tap}$ localizes the subcomponent in time and space (at scales $L_{t_p}$, $L_{x_p}$ and $L_{y_p}$, respectively) as geostrophy has local extension of covariances. It is expressed as:

$$f_{tap}(\delta x, \delta y, \delta t) = \begin{cases} \cos\left(\dfrac{\pi}{2}\delta x\right)\cos\left(\dfrac{\pi}{2}\delta y\right)\cos\left(\dfrac{\pi}{2}\delta y\right), & for\ \big(|\delta x|, |\delta y|, |\delta y| < (1,1,1)\big) \\ 0, & elsewhere \end{cases} \qquad (A17)$$

In practice, $L_{x_p}$ and $L_{y_p}$ will be set to 1.5 the wavelength of element p and $L_{t_p}$ to the decorrelation time scale. Then, the same element p of the decomposition has also an expression in geostrophic current (through the geostrophic relation Equation A15) written in the $\Gamma_{1,u}$ and $\Gamma_{1,v}$ matrices:

$$\begin{cases} \Gamma_{1,u}[i,p] = -\dfrac{g}{f_c}\dfrac{\partial \Gamma_{1,h}[i,p]}{\partial y_i} \\[2mm] \Gamma_{1,v}[i,p] = \dfrac{g}{f_c}\dfrac{\partial \Gamma_{1,h}[i,p]}{\partial x_i} \end{cases} \qquad (A18)$$

The whole time-space domain is paved with similar subcomponents, along coordinates $(x_p, y_p, t_p)$ for wavelengths between 80 km and 900 km spanning in all directions of the plan. The ensemble can be seen as a wavelet basis. Finally, each subcomponent p is assigned an expected variance in the $Q_1$ matrix, consistent with the power spectrum observed from altimetry at the corresponding wavelength with isotropy assumption.

*Equatorial waves component*

We define here the gridded variables $H_2$ and $H_3$ to resolve TIW and Poincaré waves, respectively, and we consider no contributions of the equatorial wave components on the geostrophic currents, therefore the corresponding gridded geostrophic current fields $(U_2, V_2)$ and $(U_3, V_3)$ writes: $U_2=U_3=0$, $V_2=V_3=0$. The reduced state is represented in the time-space domain by the following $\Gamma_{2,h}$ and $\Gamma_{3,h}$ matrix:


$$\Gamma_{2,h}[i,p] = \cos\left(\omega_{2,t,p}(t_i - t_p) - k_{2,x,p}(x_i - x_p)\right) * f_{tap}\left(\frac{x_i - x_p}{L_{2,x_p}}, \frac{y_i - y_p}{L_{2,y_p}}, \frac{t_i - t_p}{L_{2,t_p}}\right) \qquad (A19)$$

$$\Gamma_{3,h}[i,p] = \cos\left(\omega_{3,t,p}(t_i - t_p) - k_{3,x,p}(x_i - x_p)\right) * f_{tap}\left(\frac{x_i - x_p}{L_{3,x_p}}, \frac{y_i - y_p}{L_{3,y_p}}, \frac{t_i - t_p}{L_{3,t_p}}\right) \qquad (A20)$$

where $k_{2,x,p}$ and $k_{3,x,p}$ refer to the zonal wavenumber, and $\omega_{2,t,p}$ and $\omega_{3,t,p}$ are the frequency which satisfies the dispersion relation of the wave component (Matsuno, 1966), e.g.:


$$\begin{cases} \omega_{2,t,p} = c_2.k_{2,x,p} \text{ for the TIW, } c_2 = -0.5 \, m.s^{-1} \\ \omega_{3,t,p} = \sqrt{k_{3,k,p}^2.c_3^2 + \beta.c_3.(2.n+1)} \text{ for the Poincaré waves, } c_3 = \pm 2.8 \, m.s^{-1} \end{cases} \qquad (A21)$$

With $c_2$ and $c_3$ the wave propagation speed (the sign indicating the direction of propagation, negative for westward, positive for eastward), $\beta$ the meridional gradient of the Coriolis frequency $f_c$, and $n = 1,2,3 \ldots$.

In the present study, we chose $L_{2,t_p} = 20 \, days$, $L_{2,x_p} = 500 \, km$ and $L_{2,y_p} = 300 \, km$ for the TIW component; $L_{3,t_p} = 5 \, days$, $L_{3,x_p} = 1000 \, km$ and $L_{3,y_p} = 300 \, km$ for equatorial Poincaré wave component. As for the geostrophy component, the function $f_{tap}$ localizes the subcomponent in time and space (at scales $L_{t_p}$, $L_{x_p}$ and $L_{y_p}$, respectively).

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
