# Peer review of "Improved global sea surface height and currents maps from remote sensing and in situ observations"

_Earth System Science Data, 2022_

## Author Comment (AC1)

Reply to Referee #1:

**Authors response (AR): We would like to thank referee #1 for his/her review and for the suggestions to improve the manuscript. We respond below to the referee's comments.**

General Comments

This paper begins by presenting a new gridding method for producing maps of currents and sea surface height by combining data from altimeters and measurements from drifting buoys.

The method was already proposed in a previous work published by one of the authors of this paper and tested using an Observing System Simulation Experiment (OSSE) and Observing System Experiment (OSE). Here the method is applied for the first time to real data and the results appear to be quite interesting.

In its current form, the article also includes a very long description of the mapping method that has already been published, which, at the same time, is also too short for readers unfamiliar with the mathematical details of the discussion. My suggestion is to move section 2.2 (methods) to an appendix leaving in the main text only a qualitative introduction to the two gridding methods. This will also give the opportunity to add some missing information, such as, for example, justify the choice of covariance function or the limit to 1000 observations, which I assume is the result of several trials.

**AR: As suggested by the reviewers, we moved the technical description of the DUACS and MIOST mapping approach to an Appendix section, leaving in the main text our motivation for comparing the DUACS and MIOST approaches. Section 2.2 has been rewritten.**

The major merit of this paper is to propose the combined use of al the useful and available data (altimeters and drifters) to obtain an improved product for the global ocean circulation also in view of the future missions based on large swath technologies. Even if the actual improvement of the currents and seal level is not very impressive, I am convinced that the method and the strategy of using data form very different platforms is more than promising. In this sense, I would also be curious to know how far this new interpolation method is from being used in an operational context such as CMEMS.

Overall, I would say that it is a good paper that deserves to be published doing some revisions as suggested in this review

Recommendation: minor to major revisions

Specific Comments

Section 2.1, table 1: date interval in the table "20160115-20200630" please put a space or any other kind of separator between year month and day (this applies for all the dates in the paper). In the same table also add "degrees" in the spatial coverage line. And also define AOML.

**AR: We corrected it in the new version of the manuscript**

Section 2.1.1 line 79-80: Add a reference.

**AR: We added "(see Taburet et al., 2021, for the reference associated to each mission corrections)" in the updated version of the manuscript.**

Section 2.1.1 figure 1: How many altimeters are included in these 7 days period?

**AR: We mentioned the number of altimeters in the updated version of the manuscript**

Section 2.1.2, line 89: The reference to Prandi et al. is not in the references section. This is not the only missed reference, please check the reference section.

 **AR: This has been corrected in the updated version.**

Section 2.1.2: probably some of the readers might be interested in understanding how the altimeter can measure sea level in ice-covered areas. Can you add few words about this?

**AR: Sentences have been added in the introduction of the section**

Section 2.1.3: Really a lot of model-based corrections!  How much better is this geostrophic estimate than using geostrophic currents directly derived from models from their sea level elevation estimates (when produced)?

**AR: The experimental design proposed in this study was mainly motivated by using only observation as input datasets.  We are aware that each dataset (from model or observation) has its limitations (residual error in observations or models, constrained scales in models, …). Here, we wanted to keep/constrain the part of the geostrophy signal that is consistent between two sources of observations: the altimeter data and the insitu drifters' data.**

Section 2.1.3, figure 3: no drifters in the Mediterranean Sea in 2019?

**AR: There is no drifters in the Med. Sea because the Ekman correction is not available for the basin. Consequently, we didn't include the available drifters in the analysis. We added a sentence about it in the description of the drifter's dataset: "The Ekman component is not available in the Mediterranean basin, so there is not drifter used in this region for the study."**

Section 2.2.1, line 143: Ducet et al 2000 in not in the reference section

**AR: Corrected in the updated version**

Section 2.2.1, it would be interesting to see the covariance function. Also, Arhan and Colin de Verdière (1985) in not in the reference list. Definitely the reference section needs to be carefully reviewed!

**AR: We added the reference and the formulae of the Ahran and Colin de Verdiere covariance model.**

Section 2.2.1, line 176: "(in this study N=3". The second parathesis is missing.

**AR: Parathesis is removed**

Section 2.2.1, lines 221-222:  "the result strongly relies on the choice of covariance models". Once again, if this choice is so important, I suggest to show your choice.

**AR: Examples of covariance models for geostrophy and equatorial waves are provided in section 2.2.2 of the manuscript.**

Section 3.1, line 290: "geostrophic current anomaly data from AOML drifter database" How "geostrophic currents" are computed from drifter (by the way lagrangian) velocities?

**AR: the geostrophic current computation is mentioned in the section 2.1.3 with equations 2 and 3, and it is also described in the section.**

Section 3.1, line 293-294: what is the criterion used to select the 20% to be excluded?

**AR: No investigation was conducted to find specific criteria to justify the choice to exclude 20% of drifters. Drifters were excluded randomly. We needed to exclude enough drifters to be able to account for enough data in the mapping while having enough independent data for validation.**

Section 3.2, line 315: the mentioned "geostrophic velocity errors" refers to the intensity or to a specific component?

**AR: It refers to the zonal and meridional component: we reformulated as follows: "The similar statistical analysis can also be performed on the**

**geostrophic velocity errors Uerror = Umap- Udrifter, for the zonal component, and Verror = Vmap - Vdrifter, for the meridional component".**

Section 3.2, lines 333-334: The criterion used to determine the effective resolution is not justified. If not an explanation at least a reference is needed. Moreover, can the slope of the PSD contribute to determine the effective resolution?

**AR: We added a reference: " As in Ballarotta et al. (2019), the effective resolution is then given by the wavelength λs where the SNR(λs) is 2 (Equation 25), i.e., the wavelength where the SSHerror is two times lower than the signal SSHalongtrack."**

Figure 6: Why not show the two variance maps as well?

**AR: We did not show the two variance maps (MIOST and DUACS) because they are relatively close at 10% as shown in the manuscript figure. We propose to update the figure by including the two variance maps and their differences.**

Table 4:  Perhaps you need more digits to appreciate differences of less than 1%? Is that reasonable? Why can you say 0.0% for the Arctic and -0.8 for the equatorial belt when you read the same numbers in columns 2 and 3? Of course, this question applies also for the other tables.

**AR: It is right, more digits are needed to appreciate the differences less than 1%. We corrected all the tables in using numbers with more digits and using cm unit instead of meter.**

Section 4.2.1, Geostrophic current quality: "Overall, MIOST surface velocities are slightly closer to drifter velocities than the DUACS surface velocities." can it be said that MIOST is closer to the drifters also because it applies a kind of assimilation of them?

**AR: This question is difficult to answer. By construction, the MIOST maps will be closer to the 20% drifter's dataset used for the validation because the physical content of the assimilated drifters is somehow "injected" into the MIOST maps whereas the DUACS maps do not assimilate drifter data. On the other hand, the comparison of MIOST allsat-1 and DUACS allsat-1 experiment (provided in the manuscript) which do not assimilate drifters shows that the MIOST maps remain closer to the actual drifter data than the DUACS maps. So even without drifter information the MIOST surface velocities are slightly closer to drifter velocities than the DUACS surface velocities**

Table 6: It would probably be interesting to show the error for velocity intensity as well.

**AR: The table on regionally averaged mapping error variance and gain/reduction of error variance for the surface currents between experiment MIOST allsat-1 and MIOST allsat-1 80% drifters + equatorial waves+ L3 arctic is shown in Figure 7**

Section 5: Ubelmann et al (2020, 2021): 2020 or 2016? Once again control the reference section.

**AR: References are corrected in the updated version**

---

## Author Comment (AC2)

Reply to Referee #2:

**Authors response (AR): We would like to thank referee #2 for his/her review and for the suggestions to improve the manuscript. We respond below to the referee's comments.**

This manuscript presents a new gridded sea surface height and current dataset produced by combining observations from nadir altimeters and drifting buoys. The application of the MIOST solution is extended to the simultaneous mapping of equatorial waves and mesoscale circulation from real observations. These results pave the way for the exploration of new types of ocean signals that may eventually be mapped from remote sensing and in situ observations.

In the introduction section, it is better to review and summarize different global gridded sea surface current datasets that already exist. Also, further highlight the differences and advantages between this data set and other previous data sets.

**AR: We have considered the recommendations suggested by the referee by adding references to existing global gridded sea surface current datasets. In addition to the discussion on the limitation of surface variabilities resolved by DUACS maps, we have extended the summary & conclusion section with a paragraph presenting existing datasets that can be used as an alternative to DUACS/MIOST maps, from line 483:**

**"It is also worth mentioning that several other global gridded products exist as an alternative to the DUACS/MIOST products which provide only the geostrophic part of the surface current. Examples of these other products that provide a broader spectrum of ocean surface current variability (e.g., the total surface currents) include, 1) the Copernicus GLORYS12v1 global ocean reanalysis (Lellouche et al., 2018; https://doi.org/10.48670/moi-00021), 2) the Copernicus GLOBCURRENT product (Rio et al., 2014; https://doi.org/10.48670/moi-00050), 3) the OSCAR product (Dohan, 2021; https://doi.org/10.5067/OSCAR-25I20) distributed by the NASA-JPL Distributed Physical Oceanography Active Archive Center (PO. DAAC)."**

---

## Author Comment (AC3)

**Authors response (AR): We would like to thank referee #3 for his/her review and for the suggestions to improve the manuscript. We respond below to the referee's comments.**

Review of the MS by Ballarotta et al. (essd-2022-181)

The MS makes a significant step in reconstruction of gridded global sea level and surface currents over extended range of scales, based on fusing the data from different sources. While Level-3 altimetry is the main data source, known problems related to anisotropy of sampling density in along-track and cross-track directions are gradually solved by inclusion of independent surface drifter data. From methodical point of view, classical optimal interpolation scheme, used in the operational DUACS setup, is extended in the experimental MIOST method. The new method allows inclusion of theoretical knowledge of the involved processes (i.e. geostrophy, divergent and high-frequency ageostrophy, dispersion relations of basic wave types etc) based on the wavelet decomposition of original physical state vectors into scale-dependent components in the time-space domain. Such an approach is promising.

Practical study is made on the basis of data from three sources: CMEMS Global Altimeter SLA products, experimental Arctic leads Altimeter SLA products, and drifter trajectories related to the AOML Drifters' geostrophic velocity product. Drifter data were not used in the equatorial strip.

While overall material of the MS is clear, interesting and worth publishing, I give some comments which may be taken into account when preparing the final MS.

**AR: Reply to point 1 to 5: following the recommendations of the reviewers, we have re-written section 2.2.2. We moved the technical description of the DUACS and MIOST to an Appendix section, leaving in the main text a short description of each method and our motivation for comparing the DUACS and MIOST approaches.**

1. The MS involves a number of issues from physical oceanography, but their presentation is rather fragmentary. There could be an outline of the scales and main features of the processes that are considered in the methods and results, perhaps in the introduction or in the beginning of sub-sections of section 2.2.2 (see the next comment). For example, it could be noted, that slow Rossby waves are solved already within geostrophy, but faster equatorial waves (TIW and Poincaré) benefit from direct inclusion of their dispersion relations.

**AR: We modified section 2.2.2 accordingly.  The limitations of the DUACS mapping method are presented from line 154.**

2. The section "2.2.2 A multiscale & multivariate mapping approach" is too technical.

The section start with noting the oversampling problem (lines 168-173) is not helpful and could be skipped. The section could briefly summarize the motivation by Ubelmann et al. (2021) written in the Plain Language Summary. The list in lines 175-179 /1) geostrophy and equatorial waves: 2) Tropical Instability Waves (TIW) and 3) Poincaré waves/ could be elaborated and explained; it could be also harmonized with caption of Figure 7 and equations 20, 21. I propose that in the beginning of each approach of inclusion of analytical expressions, description of specific oceanographic processes could be given together with references to the basic literature.

**AR: The technical part is now moved to an Appendix. We summarized our motivation for testing the MIOST mapping approach from line 154.**

3. Presentation of matrix-vector operations in lines 178-222 is exactly copied from the paper by Ubelmann et al. (2021). Such repetition is not necessary. The section could be condensed and explained for the user who is not intending to make own implementation or development of the method, but rather interested to understand the basic steps behind the new gridded data sets. It seems that the key equations are 9, 10 and 15. Regarding (10), it could be explicitly written that index k (1...3) presents different type of physics as stated earlier. Type of the wavelet decomposition (line 189, but also 240, 253) is not clear and could be explained

**AR: Technical description of the MIOST approach is moved to the Appendix.**

4. The section of geostrophy component in lines 225-255 is again a direct copy from Ubelmann et al. (2021). It is necessary for the next parts of the MS. Still, it can be modified for better readability. For example, more physics like quasi-geostrophic motions, including Rossby waves (later referenced in line 373), could be noted.

**AR: The geostrophy component is used in this study and we find it necessary is keep the formulation provided by Ubelmann et al; 2021 since it is also used for introducing the equatorial waves components. The description of the geostrophy component is moved to the Appendix.**

5. Compared to Ubelmann et al. (2021), the MS presents new approach for two types of equatorial waves. Since earlier in lines 175-179 these waves were indexed k=2 and k=3, then dispersion relations Eq (21) should be split into two, with appropriate indexes. Literature references to the dispersion relations should be given.

**AR: The technical description of the Equatorial waves is moved in the Appendix. We added indexes 2 and 3 to refer to the equatorial wave's components for TIW and Poincaré waves, respectively. We also added reference to the dispersion relations.**

6. Reasons for excluding the drifter data in the equatorial zone (lines 294-296) are not clear; are the drifter data too noisy to evaluate equatorial waves, or some other reasons.

**AR: We excluded the drifters' data in the equatorial band because we wanted to isolate only the contribution of the equatorial wave modes in our analysis. We mention it in the revised version of the manuscript from line 215: "The Saral/Altika dataset (over open ocean region) and the remaining 20% of the drifter trajectories were here excluded from the mapping to perform independent assessments. Note that for these specific maps, drifter trajectories between -10°S and 10°N (e.g., Figure 3) were also excluded to evaluate only the impact of the equatorial wave's mode in this region."**

7. Description of experiments (lines 296-299) does not agree with the Table 3; data from Copernicus Marine Service referred in the text are not in the table.

**AR: We re-phrase the paragraph on the description of the experiments (from line 206) as: "Specific maps were also made to quantitatively assess the quality of these MIOST products. Table 3 summarises the list of experiments conducted in this study, indicating the input data used in the mapping and the physical content of the maps.**

**DUACS allsat-1 and MIOST allsat-1 experiments focus on the geostrophic variability. These SSH maps were produced from six altimeters (Jason-3, Cryosat-2, Sentinel-3A, Sentinel-3B, Haiyang-2A, Haiyang-2B) for the period 2019-01-01 to 2019-12-31, excluding one altimeter (Saral/Altika, over open ocean region) from the mapping to perform independent assessments. The MIOST allsat-1 80% drifters + equatorial waves+ L3 arctic experiment focuses on the geostrophic and equatorial waves variabilities. This experiment is based on 1) 80% of the drifter data, 2) the six altimeters previously mentioned over ocean and 3) leads altimeter observations. The Saral/Altika dataset (over open ocean region) and the remaining 20% of the drifter trajectories were here excluded from the mapping to perform independent assessments. Note that for these specific maps, drifter trajectories between -10°S and 10°N (e.g., Figure 3) were also excluded to evaluate only the impact of the equatorial wave's mode in this region. "**

8. Vorticity results are presented without any explanation (line 344, Fig. 5). Why we need them, are we interested in eddies etc?

**AR: We removed the vorticity maps in the updated version of the manuscript since we do not discuss about it**

9. Theoretical dispersion curves in Fig. 7 are not labeled. Their main features are not explained and/or referenced in the text.

**AR: We updated the figure with notation for each curve in the updated version of the manuscript**

Technical issues

a) The title starts with "Improved global sea surface height and currents maps...". Although comparison with the existing sea level maps (E.U. Copernicus Marine Service (product reference SEALEVEL_GLO_PHY_L3_MY_008_062) show some improvement during the 4 years test period, in my understanding it is not yet finally clear whether the new method is also an improved method in statistical sense.

**AR: As mentioned in the conclusion, we believe that the products based on the MIOST approach are improved compared with DUACS approach from several qualitative and quantitative aspects: "...Arctic leads SSH observations allow to significantly improve the monitoring coverage in this remote region. (...) Drifters' observations ... mainly contribute to reduce mapping errors in regions of intense dynamics where the temporal sampling must be accurate enough to properly map the rapid mesoscale dynamics. (...) we explored and validated the possibility of improving the content of altimetry maps by simultaneously estimating the ocean mesoscale circulations as well as the equatorial wave dynamics associated to the Tropical Instability and Poincaré Waves, and (...) show that mapping these ocean surface variabilities from altimeter observations broadens the spectrum of mappable space-time scales and reduces mapping errors by almost 20% locally relative to independent data, primarily in the equatorial Pacific and Atlantic basins. " Also note that the statistical assessment in section 4.2.2 focuses on the improved mapping associated with equatorial waves and drifters with MIOST.**

b) Lines 22-23 state that "this new product is proposed against the DUACS operational product distributed in the Copernicus Marine Service." The wording may create a feeling of contradiction, but in essence, the new method is meant to be used in further development of CMEMS.

**AR: We re-phrase as: "A quality assessment of this new product is proposed with regard to an operational product distributed in the Copernicus Marine Service"**

c) Line 28: "effective resolution" remains unclear in the abstract, although it is well presented in the main text.

**AR: We propose to change "effective resolution" to "resolved scales"**

d) There is a number of unexplained abbreviations like AVISO, DUACS, MIOST, AOML, SWOT.

**AR: We corrected it in the updated version of the manuscript**

e) The name DUACS in the section 2.2.1 heading (line 140) is not informative. The section presents optimal interpolation used in the operational setup.

**AR: We change section heading as: The Optimal Interpolation (DUACS mapping approach). Note that we mode the technical description of the DUACS mapping method in the Appendix.**

f) Line 256 should have reference to Fig. 4a and 4c (not the whole Fig. 4), since westward propagating graphs (Fig. 4b and 4d) are introduced in the next section.

**AR: We corrected it in the updated version of the manuscript**

g) Lines 376-378 (caption of Fig. 7) lacks the notation, which curves correspond to Kelvin, Yanai, Rossby and Poincaré waves.

**AR: We updated the figure with notation for each curve in the updated version of the manuscript**

h) There should be space between the number and the unit (lines 386-389, 412, 414 and so on).

**AR: We corrected it in the updated version of the manuscript**

i) Naming of the experiments should be unified throughout the MS. There are EXP01 to EXP03 listed in Table 3, but other names are given in the captions of Figs. 8-14.

**AR: We corrected it in the updated version of the manuscript: we unified the naming of the experiments explicitly as: *DUACS all-1*, *MIOST allsat-1*, and *MIOST allsat-1 + equatorial waves + L3 arctic***

j) Reference list could be extended to include basic papers in physical oceanography, relevant to the altimetry development issues.

**AR: Several physical oceanography papers have been included in the updated version of the manuscript, in particular reference paper related to equatorial waves dynamics.**

k) References should be ordered according to the journal rules (alphabetically), presently there are flaws. Paper by Le Guillou lacks reference to the publication year.

**AR: We corrected it in the updated version of the manuscript**